# A Data-Augmentation Is Worth A Thousand Samples: Analytical Moments And Sampling-Free Training

**Randall Balestriero**
Meta AI Research, FAIR
NYC, USA
`rbalestriero@meta.com`

**Ishan Misra**
Meta AI Research, FAIR
NYC, USA
`imisra@meta.com`

**Yann LeCun**
Meta AI Research, FAIR, NYU
NYC, USA
`ylecun@meta.com`

## Abstract

Data-Augmentation (DA) is known to improve performance across tasks and datasets. We propose a method to theoretically analyze the effect of DA and study questions such as: how many augmented samples are needed to correctly estimate the information encoded by that DA? How does the augmentation policy impact the final parameters of a model? We derive several quantities in close-form, such as the expectation and variance of an image, loss, and model's output under a given DA distribution. Up to our knowledge, we obtain the first explicit regularizer that corresponds to using DA during training for non-trivial transformations such as affine transformations, color jittering, or Gaussian blur. Those derivations open new avenues to quantify the benefits and limitations of DA. For example, given a loss at hand, we find that common DAs require tens of thousands of samples for the loss to be correctly estimated and for the model training to converge. We then show that for a training loss to have reduced variance under DA sampling, the model's saliency map (gradient of the loss with respect to the model's input) must align with the smallest eigenvector of the sample's covariance matrix under the considered DA augmentation; this is exactly the quantity estimated and regularized by TangentProp. Those findings also hint at a possible explanation on why models tend to shift their focus from edges to textures when specific DAs are employed.

## 1 Introduction

Data Augmentation (DA) is a prevalent technique in training deep learning models [1, 2, 3]. These Deep Networks (DNs) models $f_\gamma$, governed by some parameters $\gamma \in \Gamma$, are trained on the train set and expected to generalize to unseen samples (test set). To combat the tendency of DNs to overfit [4], producing a large train-test performance gap, regularization and in particular DA is heavily employed, amongst other mechanisms such as weight-decay [5, 6]. The benefit of DA over those alternatives is that defining input transformations that preserve the semantics of their inputs is a relatively simple task, at least in computer vision or acoustic processing [7, 8, 9]. Furthermore, if the DA is well designed and rich enough, it can effectively bring the number of non-trivial training samples close to the theoretical limits ensuring that any performance gap between the train and test set vanishes [10] as empirically observed in various scenarios [11, 12]. In fact, DA has proven so useful that novel training methods such as self-supervised learning (SSL) entirely rely on DA [13, 14] to learn meaningful data representations.

Despite its empirical effectiveness, our understanding of DA has many open questions, three of which we propose to study: **(a)** how do different DAs impact the model's parameters during training?; **(b)** how sample-efficient is the DA sampling, i.e., how many DA samples a model must observe to converge?; and **(c)** how sensitive is a loss/model to the DA sampling and how this variance evolves during training as a function of the model's ability to minimize the loss at hand, and as a

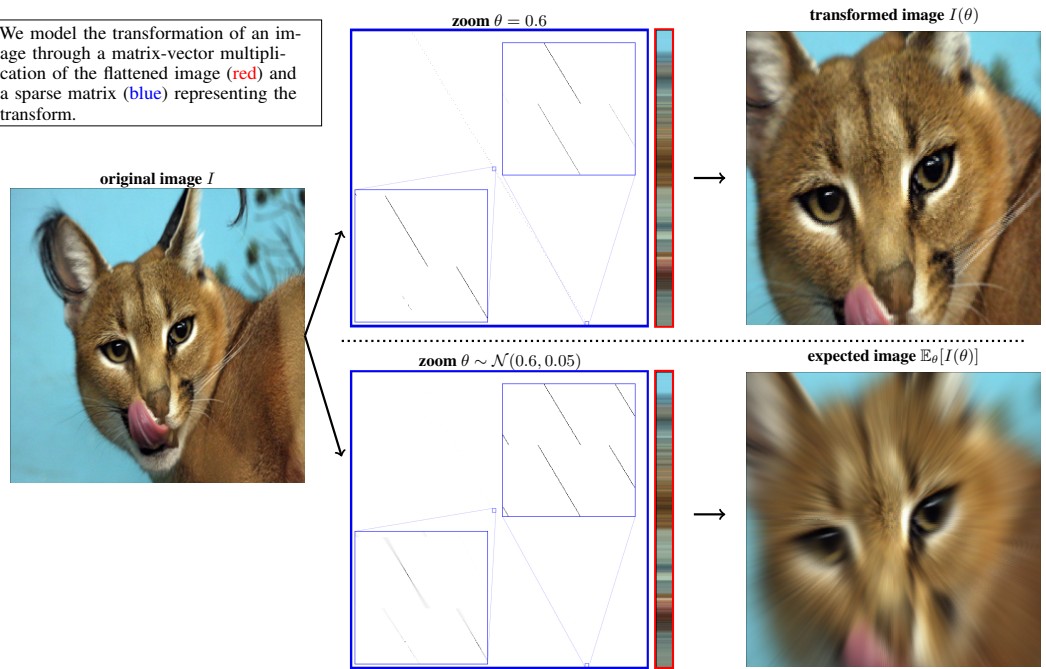

Figure 1: We propose a novel way to express Data Augmentation (DA) analytically that allows us to understand the impact of DA on the learned parameters of a model and quantify DA's sample efficiency. This allows us to compute the analytical expectation and variance of the transformed data (or any function computed with it) in close-form with respect to the transformation parameters.

function of the model's parameters? Our goal is to analytically derive some preliminary answers and insights into those three areas enabled by a novel image transformation operator that we introduce in Section 3.2 coined Data-Space Transform (DST). DST employs a mathematical formulation of input transformations akin to the one of [15] although slightly more general as we do not require the DA to form a group. We study the zoom/translation/shearing/rotation DAs in the main text and Gaussian blur, grayscale, color jitter and random crop in Appendix C; code is available[1]. We summarize our contributions below:

(a) we derive the analytical first order moments of augmented samples, and of the losses employing augmented samples (Section 3.3), effectively providing us with the explicit regularizer induced by each DA (Section 3.4)
(b) we quantify the number of DA samples that are required for a model/loss to obtain a correct estimate of the information conveyed by that DA (Section 4.1)
(c) we derive the sensitivity, i.e. variance, of a given loss and model under a DA policy (Section 4.2) leading us to rediscover from first principles a popular deep network regularizer: TangentProp, as being the natural regularization to employ to minimize the loss variance (Section 4.3).

**Upshot of gained insights:** **(a)** will show us that the explicit DA regularizer corresponds to a generalized Tikhonov regularizer that depends on each sample's covariance matrix largest eigenvectors; the kernel space of the model's Jacobian matrix aligns with the data manifold tangent space as modeled by the DA; **(b)** will show us that the number of augmented samples required for a loss/model to correctly estimate the information provided by a DA of a single sample is on the order of $10^4$. Even when combining the information of thousands of samples, we find that the entire train set must be augmented at least $50\times$ for a DA policy to be correctly learned by a model; lastly **(c)** will quantify the loss variance as a simple function of the model's Jacobian matrix, and the eigenvectors of the augmented sample variance matrix. Regardless of the model or task-at-hand, the loss sensitivity to random DA sampling goes down as the kernel of a model's Jacobian matrix aligns with the principal directions of the data manifold tangent space; as a by-product, we recover TangentProp as the natural regularizer minimizing the loss sensitivity to DA sampling.

---

[1] https://github.com/facebookresearch/analytical_augmentation_moments

**Limitations:** all of our image-space results e.g. expectation and variance of images under DA are exact, no approximation is employed. Results on a model's output/loss are provided both in the case of a linear model and of a nonlinear model. For the former, results are exact, for the latter a third-order Taylor approximations is employed which is common and has been shown to provide accurate enough approximations to safely rely on the insights/quantities obtained e.g. see Sec. A.2 of [16]. All the proofs and implementation details are provided in the appendix. We also emphasize that our primary goal is to provide a theoretical understanding of DA, to quantify its efficiency and impact onto learned models and to derive and visualize its explicit regularizer; we leave the search for tractable alternatives to DA based on those insights for future work.

## 2  Background

**Existing Explicit Regularizers From Data-Augmentation.**    It is widely accepted that data-augmentation (DA) regularizes a model towards the transformations that are modeled [5, 17], and that this regularization impacts performances significantly and positively, possibly as much as the regularization offered by the choice of DN architecture [18] and optimizer [19].

To gain precious insights into the impact of DA onto the learned functional $f_\gamma$, the most common strategy is to derive the *explicit* regularizer that directly acts upon $f_\gamma$ in the same manner as if one were to use DA during training. This explicit derivation is however challenging and so far has been limited to DA strategies such as additive white noise or multiplicative binary noise applied identically and independently throughout the image and/or feature maps, as with dropout [20, 21]. In those settings, various works have studied in the linear regime the relation between such DA and its equivalence to using Tikhonov regularization[22] or weight decay [23] as in $\min_{\boldsymbol{W}} \sum_{n=1}^{N} \mathbb{E}_{\boldsymbol{\epsilon}\sim\mathcal{N}(0,\sigma)} \left[\|\boldsymbol{y}_n - \boldsymbol{W}(\boldsymbol{x}_n + \boldsymbol{\epsilon})\|_2^2\right] = \min_{\boldsymbol{W}} \sum_{n=1}^{N} \|\boldsymbol{y}_n - \boldsymbol{W}\boldsymbol{x}_n\|_2^2 + \lambda(\sigma)\|\boldsymbol{W}\|_F^2$ [24]. More recently, [25, 26] extended the case of additive white noise to nonlinear models and concluded that additive white noise DA corresponds to adding an explicit Frobenius norm regularization onto the Jacobian of $f_\theta$ evaluated at each data sample.

Going to more involved DA strategies e.g. translations or zooms of the input images is challenging and has so far only been studied from an empirical perspective. For example, [27, 28, 29] performed thorough ablation studies on the interplay between DAs and a collection of known explicit regularizers to find correlations between them. It was concluded that weight-decay (the explicit regularizer of additive white noise) does not relate to those more advanced DAs. This also led other studies to suggest that norm-based regularization might be insufficient to describe the implicit regularization of DAs involving advanced image transformations [30]. We debunk this last claim in Sec. 3.3.

**Coordinate Space Transformation.**    Throughout this paper, we will consider a two-dimensional image $I(x, y)$ to be at least square-integrable $I \in L^2(\mathbb{R}^2)$ [31]. Multi-channel images are dealt with by treating each channel as its own single-channel image, as commonly done in practice [1]. As we are interested in practical cases, we will often assume that $I$ has compact support e.g. has nonzero values only within a bounded domain such as $[0, 1]^2$. Visualizing this image thus corresponds to displaying the sampled values of $I$ on a regular grid (pixel positions) of $[0, 1]^2$ [32]. The most common formulation to apply a transformation on the image $I$ to obtain the transformed image $T$ is to transform the image coordinates [33, 34, 35]. That is, a mapping $t : \mathbb{R}^2 \mapsto \mathbb{R}^2$ describes what coordinate $t(u, v)$ of the original image $I$ maps to the coordinate $u, v$ of the transformed image $T$ as in

$$T(u, v) = I(t(u, v)). \tag{1}$$

This mapping $t$ often comes with some parameters $\theta$ governing the underlying transformation as in

$$t_\theta(x,y) = [x - \theta_1, y - \theta_2]^T, \quad t_\theta(x,y) = \begin{bmatrix} \cos(\theta) & -\sin(\theta) \\ \sin(\theta) & \cos(\theta) \end{bmatrix} \begin{bmatrix} x \\ y \end{bmatrix}, \quad t_\theta(x,y) = [\theta_1 x, \theta_2 y]^T, \tag{2}$$

for translation, rotation and zoom respectively. We provide a visual depiction of the zoom transformation applied in coordinate space in Fig. 6, in the appendix.

The formulation of Eq. 1 has two key benefits. First, it allows a simple and intuitive design of $t$ to obtain novel transformations. Second, it is computationally efficient as the coordinate-space of images are 2/3-dimensional. Those benefits have led to e.g. the Spatial Transformer Network [36]. On the other hand, Eq. 1 has one major drawback for our purpose: the exact moments of the transformed image under random $\theta$ parameters are not tractable due to the composition of $t$ with the nonlinear mapping $I$. And as it will become clear in Section 3.1, those quantities are needed to derive DAs' explicit regularizers.

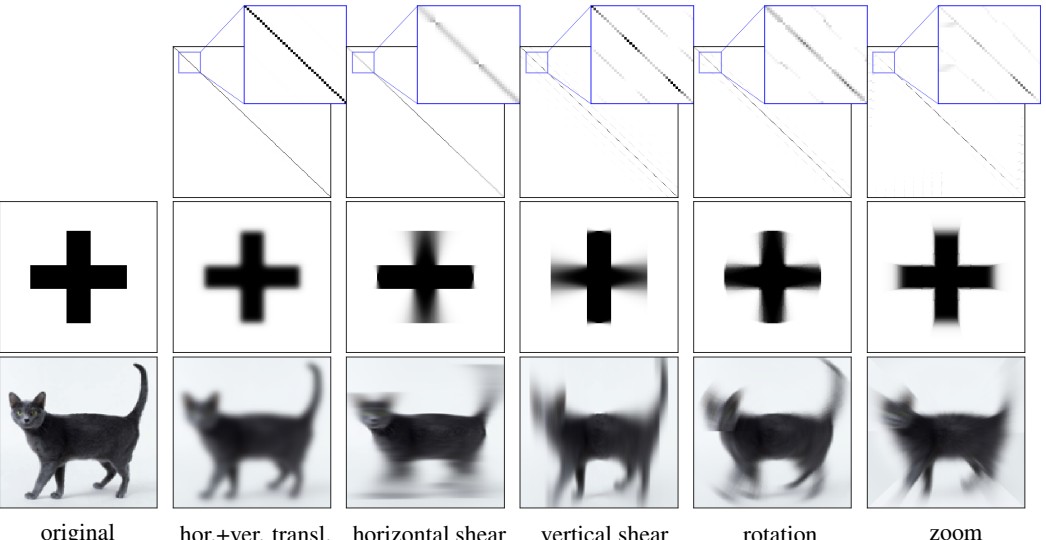

| original | hor.+ver. transl. | horizontal shear | vertical shear | rotation | zoom |

Figure 2: **Top row:** sparse matrix $\mathbb{E}_\theta[\boldsymbol{M}(\theta)]$ producing the expected image via $\mathbb{E}_\theta[\boldsymbol{t}(\theta)] = \mathbb{E}_\theta[\boldsymbol{M}(\theta)]\boldsymbol{x}$ (recall Eq. (7) and the discussion below Theorem 3.3), here with DA distributions of $\mathcal{N}(0, 0.04) \otimes \mathcal{N}(0, 0.04)$, $\mathcal{N}(0, 0.2)$, $\mathcal{N}(0, 0.1)$ and $\mathcal{N}(1, 0.1)$ for each column respectively. This *sparse* matrix is independent of the image and derived in close-form (Theorem 3.3). **Middle & Bottom row:** are two images that depict for each column the "expected image" under each DA. Estimating those images (and any loss employing them) only through sampling the DA augmentations would require tens of thousands of samples (see Fig. 4). *Gaussian blur, grayscale, color jitter and random crop DA are provided in Appendix C along with discussions on how to extend our results to CutOut and MixUp.*

## 3 Analytical Moments of Transformed Images Enable Infinite Dataset Training

We first motivate this study by formulating the training process under DA sampling as doing a Monte-Carlo estimate of the true (unknown) expected loss under that DA distribution (Section 3.1). Going beyond this sampling/estimation procedures requires knowledge of the average and variance of a transformed sample, i.e. the first two centered-moments. This is our motivation to construct a novel Data-Space Transformation (DST) (Section 3.2) that will enable close-form formula for those moments (Section 3.3). From those, we will be able to remove the need to sample transformed images to train a model, by obtaining the close-form expected loss Section 3.4, under which training can be performed sampling-free.

### 3.1 Motivation: Current Data-Augmentation Training Performs Monte-Carlo Estimation

Training a model with DA consists in (i) sampling transformed images at each training iteration for each sample $\boldsymbol{x}_n$ as in $\mathcal{T}_{\theta_n}(\boldsymbol{x}_n)$ with $\theta_n \sim \boldsymbol{\theta}$ a randomly sampled DA parameter e.g. the amount of translation to apply, (ii) evaluating the loss $\mathcal{L}$ on the transformed sample/mini-batch/dataset, and (iii) using some flavor of gradient descent to update the parameters $\gamma$ of the model $f_\gamma$.

This training procedure corresponds to a one-sample Monte-Carlo (MC) estimate [37, 38] of the expected loss

$$\sum_{n=1}^N \mathbb{E}_{\boldsymbol{\theta}}\left[\mathcal{L}\left(f_\gamma(\mathcal{T}_{\boldsymbol{\theta}}(\boldsymbol{x}_n))\right)\right] \approx \sum_{n=1}^N \mathcal{L}(f_\gamma(\mathcal{T}_{\theta_n}(\boldsymbol{x}_n)), \tag{3}$$

with $\theta_n$ i.i.d samples of $\boldsymbol{\theta}$ repeatedly sampled at each epoch. In a supervised setting, the loss would also receive a per-sample target $\boldsymbol{y}_n$ as input. Although a one-sample estimate might be insufficient to apply the central limit theorem [39] and guarantee training convergence, the combination of multiple samples in mini-batch training does provide convergence in most cases. Although, methods such as Self-Supervised Learning that heavily rely on DA, tend to be more sensitive and unstable [40].

To avoid such instabilities, and to incorporate all the DA transformations of $x_n$ into the model's parameter update, one would be tempted to compute the model parameters' gradient on the expected loss (left-hand side of Eq. (3)) (see Section 3.4). Knowledge of the expected loss would also prove useful to measure the quality of the MC estimate (see Section 4.1). Note that even in the least-square setting, such approximation is employed as the expected loss under DAs is not know. Hence, one commonly obtain many DA samples, and then apply the least-square formula on this augmented training set, which is highly inefficient. Alternatively, this study will obtain the optimum least-square parameters under DA sampling (Theorem 3.4) i.e. only using the given training set, we directly obtain the parameters that would correspond to sampling infinitely many DA samples and applying the usual least-square formula on this infinite training set.

The close-form expected loss requires knowledge of the average and variance of the transformed sample $\mathcal{T}_{\boldsymbol{\theta}}(x_n)$ taken with respect to the random variable $\boldsymbol{\theta}$. Hence, we first propose to formulate a novel and tractable augmentation model (Section 3.2) that will allow us to obtain those moments analytically in Section 3.3.

### 3.2 Proposed Data-Space Transformation (DST)

Instead of altering the coordinate positions of an image, as done in the coordinate-space transformation of Eq. (1), we propose to alter the image basis functions.

Going back to the construction of functions, one easily recalls that any (image) function can be expanded into a basis as in $I(u,v) = \int I(x,y)\delta(u-x, v-y)dxdy$ with $\delta$ the usual Dirac distribution. Suppose for now that we consider a horizontal translation by a constant $\theta$. Then, one can obtain that translated image via

$$\mathcal{T}_\theta(I)(u,v) = \int I(x,y)\delta(u - x - \theta, v - y)dxdy, \tag{4}$$

hence, and crucial to our study, *we apply the transformation onto the basis functions onto which the image is evaluated, rather than onto the original image itself.* As the image is now constant with respect to the transformation parameter $\theta$, and as the basis functions have some convenient analytical forms, the derivation of the transformed images moments, under $\boldsymbol{\theta}$, will become straightforward. Because the transformed image $T$ is now obtained by combining its pixel values (recall Eq. (4)) we coin this transform as Data-Space Transform (DST), formally defined below.

**Definition 3.1** (Data-Space Transform). We define the DST of an image $I \in L^2(\mathbb{R}^2)$ producing the transformed image $\mathcal{T}_\theta(I) \in L^2(\mathbb{R}^2)$ as

$$\mathcal{T}_\theta(I)(u,v) = \int I(x,y)h_\theta(u,v,x,y)dxdy, \tag{5}$$

with $h_\theta(u,v,.,.) \in \mathbb{C}_0^\infty(\mathbb{R}^2)$ encoding the transformation.

In Definition 3.1, we only impose for $h_\theta(u,v,.,.)$ to be with compact support. In fact, one should interpret $h_\theta(u,v,.,.)$ as a distribution whose purpose is to evaluate $I$ at a desired (coordinate) position on its domain. This evaluation –depending on the form of $h_\theta(u,v,.,.)$– can extract a single pixel-value of the image $I$ at a desired location (as in Eq. (4)), or can combine multiple values e.g. with $h_\theta(u,v,.,.)$ being a bump function. Additionally, we do not impose any restriction on the invertibility of this mapping i.e. $\mathcal{T}_\theta(I)(u,v)$ can disregard parts of the original image.

**Coordinate-space transformations as DSTs.** The coordinate-space transformation and the proposed DST (Definition 3.1) act in different spaces: the image coordinates and the image pixel values, respectively. Nevertheless, this does not limit the range of transformations that can be applied to an image. The following statement provides a simple recipe to turn any already employed coordinate-space transformation into a DST.

**Proposition 3.2.** *Any coordinate-space transformation (1) using $t_\theta : \mathbb{R}^2 \mapsto \mathbb{R}^2$ can be expressed as a data-space transformation (5) by setting $h_\theta(u,v,x,y) = \delta(t_\theta(x,y) - [u,v]^T)$.*

We will focus in the main text on zoom/rotation/translation/shearing, and we propose in Appendix C the case of Gaussian blur, color jittering, random crop, along with a discussion on extending our results to CutMix [41] and MixUp [42]. Using Proposition 3.2, we obtain the DST operators $h_\theta(u,v,x,y)$

to be

$$\delta(u - x + \theta_1, v - y + \theta_2), \quad \delta(u - x - \theta_1 y, v - y - \theta_2 x), \quad \delta(u - \theta x, v - \theta y),$$
$$\delta(u - \cos(\theta)x + \sin(\theta)y, v - \sin(\theta)x - \cos(\theta)y), \tag{6}$$

for the vertical/horizontal translation, vertical/horizontal shearing, zoom and rotation respectively (compare with Eq. 2). Before focusing on the analytical moments of the DST samples, we describe how those operators are applied in a discrete setting.

**Discretized version.** We now describe how any DST of a discrete image, flattened as a vector, can be expressed as a matrix-vector product with the matrix entries depending on the employed DA and its parameter $\theta$, as we display in Figs. 1 and 2. The functional form of the DST from Eq. (5) producing the target image at a specific position $\mathcal{T}_\theta(I)(u, v)$ is linear in the original image $I$. Hence, in the discrete setting, the continuous integral over the image domain is replaced with a summation with indices based on the desired sampling/resolution of $I$, i.e. the position of the pixel spatial positions. Expressing linear operators as matrix-vector products will greatly ease our development, we will denote by $\boldsymbol{x} \in \mathbb{R}^{hw}$ the flattened $(h \times w)$ discrete images $I$. Hence, our data-space transformation, given some parameters $\theta$, takes the form of

$$\boldsymbol{t}(\theta) = \boldsymbol{M}(\theta)\boldsymbol{x}, \tag{7}$$

with $\boldsymbol{t}(\theta) \in \mathbb{R}^{hw}$ the flattened transformed image (which can be reshaped as desired) and $\boldsymbol{M}(\theta) \in \mathbb{R}^{hw \times hw}$ the matrix whose rows encode the discrete and flattened $h_\theta(u, v, ., .)$. For example, and employing a uniform grid sampling for illustration, $\boldsymbol{M}(\theta)_{i,j} = h_\theta(i//w, i\%w, j//w, j\%w)$ with $//$ representing the floor division and $\%$ the modulo operation. For the case of multi-channel images, we consider without loss of generality the application of the matrix-vector operation on each channel separately. We depict this operation along with the exact form of $\boldsymbol{M}(\theta)$ for the case of the zoom transformation in Fig. 1. A crucial property of the matrix $\boldsymbol{M}(\theta)$ lies in its sparsity, in fact, for most DAs the transformations rely on displacing pixels rather than combining them. Hence, although $\boldsymbol{M}(\theta)$'s total number of entries grow quadratically with the number of pixels in $I$, the number of nonzero entries only grows linearly with it.

### 3.3 Analytical Expectation and Variance of Transformed Images

The above construction (Definition 3.1) turns out to make the analytical form of the first two moments of an augmented sample much simpler to derive. As this derivation is at the core of our main contribution, we propose a step-by-step derivation of $\mathbb{E}_{\boldsymbol{\theta}}[\mathcal{T}_{\boldsymbol{\theta}}(I)]$ for the case of horizontal translation (recall Eq. (4)).

Let's consider again the continuous model (the discrete version is provided after Theorem 3.3). Using Fubini's theorem to switch the order of integration and recalling Eq. (5), we have that

$$\mathbb{E}_{\boldsymbol{\theta}}[\mathcal{T}_{\boldsymbol{\theta}}(I)(u, v)] = \int I(x, y)\mathbb{E}_{\boldsymbol{\theta}}[h_{\boldsymbol{\theta}}(u, v, x, y)]dxdy. \tag{8}$$

Using the definition for $h_\theta$ from Eq. (6) for horizontal translation, $\mathbb{E}_{\boldsymbol{\theta}}[h_{\boldsymbol{\theta}}(u, v, x, y)]$ becomes

$$\mathbb{E}_{\boldsymbol{\theta}}[\delta(u - x - \boldsymbol{\theta}, v - y)] = \int \delta(u - x - \theta, v - y)p(\theta)d\theta = p(u - x)\delta(v - y),$$

with $p$ the density function of $\boldsymbol{\theta}$ prescribing how the translation parameter is distributed. As a result, in this univariate translation case, the expected augmented image at coordinate $(u, v)$ is given by

$$\mathbb{E}_{\boldsymbol{\theta}}[\mathcal{T}_{\boldsymbol{\theta}}(I)(u, v)] = \int I(x, y)p(u - x)\delta(v - y)dxdy = \int I(x, v)p(u - x)dx, \tag{9}$$

which can be further simplified into $\mathbb{E}_{\boldsymbol{\theta}}[\mathcal{T}_{\boldsymbol{\theta}}(I)(., v)] = I(., v) \star p$. Hence, the expected horizontally-translated image is the convolution (on the $x$-axis only) between the original image $I$ and the univariate density function $p$. We formalize this for the transformations of Eq. (6) below.

**Theorem 3.3.** *The analytical form of $\mathbb{E}_{\boldsymbol{\theta}}[h_{\boldsymbol{\theta}}(u, v, x, y)]$, used to obtain the expected transformed image (recall Eq. (8)) is given by*

$$p(u - x, v - y) \ (\text{translation}), \quad \text{and} \quad p(u/x)\delta(u/x - v/y) \ (\text{rotation}), \tag{10}$$

*other cases and second-order moment $\mathbb{E}_{\boldsymbol{\theta}}[\mathcal{T}_{\boldsymbol{\theta}}(\boldsymbol{x}_n)\mathcal{T}_{\boldsymbol{\theta}}(\boldsymbol{x}_n)^T]$ are deferred to the proof. (Proof in Appendix F.)*

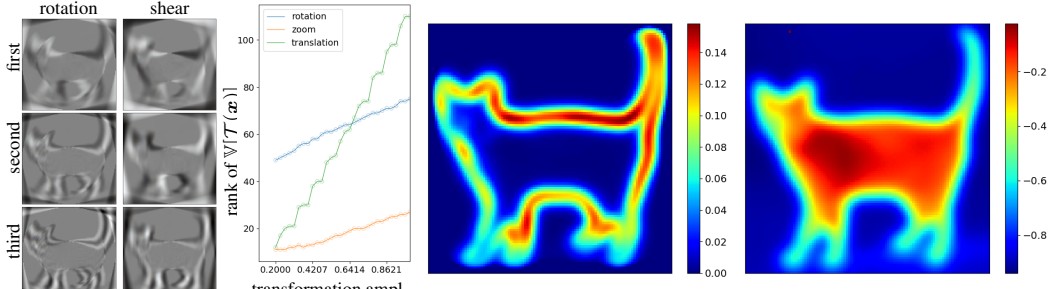

Figure 3: **First column:** top eigenvectors ($Q$ matrix in Theorem 3.4) of the sample variance matrix $\mathbb{V}[\mathcal{T}(\boldsymbol{x})]$ for rotation/shear augmentations with $\mathcal{U}(-15°, 15°)$ and $\mathcal{U}(-15°, 15°) \otimes \mathcal{U}(-15°, 15°)$ respectively. As per Theorem 4.1, training aligns the model's Jacobian matrix kernel to the largest eigenvectors of $\mathbb{V}[\mathcal{T}(\boldsymbol{x})]$ to reduce the loss variance under the respective DA sampling. **Second column:** number of nonzero eigenvalues of $\mathbb{V}[\mathcal{T}(\boldsymbol{x})]$ (nonzero elements in $\Lambda$ in Theorem 3.4) for increasing transformation amplitude (the datum dimension is $114 \times 114 \times 3$). We observe that the dimension of the subspace spanned by augmented images constraint increases linearly with the amplitude of the transformation. **Third column:** Pixel variance i.e. diagonal of $\mathbb{V}[\mathcal{T}(\boldsymbol{x})]$ reshapes as an image. **Fourth column:** pixel covariance between one background pixel and all other pixels, i.e. $700^{\text{th}}$ row of $\mathbb{V}[\mathcal{T}(\boldsymbol{x})]$ reshaped as an image for the cat of Fig. 2 under translations ($\mathcal{N}(0, 0.1) \otimes \mathcal{N}(0, 0.1)$). As per Theorem 3.4, the variance of the pixel values, seen on the left, being much higher for the edges of the cat than for its interior body/texture pushes the model to focus on the cat texture, a phenomenon empirically observed in deep networks [43, 44].

An interesting observation obtained from the above ties the expected image under random 2-dimensional translations with (2-dimensional) density $p$ to the convolution $I \star p$; this provides a new portal to study and interpret convolutions with nonnegative, sum-to-one filters $p$. Again, and as per Eq. (7), the discretized version of the expected image takes the form of $\mathbb{E}_{\boldsymbol{\theta}}[\boldsymbol{t}(\boldsymbol{\theta})] = \mathbb{E}_{\boldsymbol{\theta}}[\boldsymbol{M}(\boldsymbol{\theta})]\boldsymbol{x}$ with the entries of $\mathbb{E}_{\boldsymbol{\theta}}[\boldsymbol{M}(\boldsymbol{\theta})]$ given by discretizing Eq. (10). We depict this expected matrix for various transformations as well as their application onto two different discrete images in Fig. 2, and we now proceed on deriving the left-hand side of Eq. (3) i.e. the explicit DA regularizer.

### 3.4 The Explicit Regularizer of Data-Augmentations

To keep notations as light as possible, we first consider a linear regression model with Mean Squared Error (MSE), the nonlinear case will be studied in Section 4. In that setting, the expected loss under DA sampling (recall Eq. (3)) becomes

$$\mathcal{L} = \sum_{n=1}^{N} \mathbb{E}_{\boldsymbol{\theta}}\left[\|\boldsymbol{y}_n - \boldsymbol{W}\mathcal{T}_{\boldsymbol{\theta}}(\boldsymbol{x}_n) - \boldsymbol{b}\|_2^2\right], \tag{11}$$

with $\boldsymbol{x}_n \in \mathbb{R}^D, n = 1, \ldots, N$ the input (flattened) $n^{\text{th}}$ image $I_n$, $\boldsymbol{y}_n \in \mathbb{R}^K$ the $n^{\text{th}}$ target vector, and $\boldsymbol{W} \in \mathbb{R}^{K \times D}, \boldsymbol{b} \in \mathbb{R}^K$ the model's parameters. Recall from Section 3.1 that to learn $\boldsymbol{W}$ under DA, the current strategy consists in performing a Monte-Carlo (MC) estimation. Instead, let's derive (detailed derivation in Appendix D) the exact loss of Eq. (11) as a function of the sample mean and variance under the consider DA. We will drop the $\boldsymbol{\theta}$ subscript for clarity to obtain

$$\mathcal{L} = \sum_{n=1}^{N} \|\boldsymbol{y}_n - \boldsymbol{W}\mathbb{E}[\mathcal{T}(\boldsymbol{x}_n)] - \boldsymbol{b}\|_2^2 + \|\boldsymbol{W}\boldsymbol{Q}(\boldsymbol{x})\Lambda(\boldsymbol{x})^{\frac{1}{2}}\|_F^2, \tag{12}$$

with the spectral decomposition $\boldsymbol{Q}(\boldsymbol{x})\Lambda(\boldsymbol{x})\boldsymbol{Q}(\boldsymbol{x})^T = \mathbb{V}[\mathcal{T}(\boldsymbol{x})]$. The right term in Eq. (12) is the explicit DA regularizer. It pushes the kernel space of $\boldsymbol{W}$ to align with the largest principal directions of the data manifold tangent space, as modeled by the DA. In fact, the largest eigenvectors in $\boldsymbol{Q}(\boldsymbol{x})$ represent the principal directions of the data manifold tangent space at $\boldsymbol{x}$, as encoded via $\mathbb{V}[\mathcal{T}(\boldsymbol{x})]$.

We propose in Fig. 3 visualization of $\boldsymbol{Q}$ and $\Lambda$ for different DAs, illustrating how each DA policy impacts the model's parameter $\boldsymbol{W}$ through the regularization of Eq. (12). The knowledge of $\mathbb{E}[\mathcal{T}(\boldsymbol{x})]$ and $\mathbb{V}[\mathcal{T}(\boldsymbol{x})]$ from Theorem 3.3 finally enables to train a (linear) model on the true expected loss (Eq. (12)) as we formalize below.

**Theorem 3.4.** *Training a linear model with MSE and infinite DA sampling is equivalent to minimizing Eq.* (12) *and produces the optimal* $\boldsymbol{W}^*$ *model's parameter*

$$\boldsymbol{W}^* = \left( \sum_{n=1}^{N} (\boldsymbol{y}_n - \boldsymbol{b}) \mathbb{E}[\mathcal{T}(\boldsymbol{x}_n)]^T \right) \left( \sum_{n=1}^{N} \mathbb{E}[\mathcal{T}(\boldsymbol{x}_n)] \mathbb{E}[\mathcal{T}(\boldsymbol{x}_n)]^T + \sum_{n=1}^{N} \mathbb{V}[\mathcal{T}(\boldsymbol{x}_n)] \right)^{-1}. \quad (13)$$

Not surprisingly, whenever the DA is identity, $\mathbb{E}[\mathcal{T}(\boldsymbol{x}_n)] = \boldsymbol{x}_n, \mathbb{V}[\mathcal{T}(\boldsymbol{x}_n)] = \boldsymbol{0}$ and thus Eq. (13) recovers exactly the standard least-square solution. We visualize $\mathbb{V}[\mathcal{T}(\boldsymbol{x}_n)]$ for the translation DA in Fig. 3. The same line of result can be derived in the nonlinear setting by assuming that the DA is restricted to small transformations. In that case, one leverages a truncated Taylor approximation of the nonlinear model[2] and recovers that (local) DA applies the same regularization as in Eq. (12) but with the model's Jacobian matrix $\boldsymbol{J}f_\gamma(\boldsymbol{x}_n)$ in-place of $\boldsymbol{W}$ (more details in Section 4.3).

Given the above derivation of the exact expected loss, we can now turn to precisely measure how accurate is the MC estimate commonly used to train models under DA sampling.

## 4 Data-Augmentation Sampling Efficiency and Loss Sensitivity

In this section we study the convergence of the MC estimate (Section 4.1), and provide variance analysis of that estimate as a function of the model's Jacobian matrix and the sample variance eigenvectors (Section 4.2), concluding by the (re-)discovery of TangentProp from first principles (Section 4.3). Those results hold regardless of the type of DA employed. Additionally, the analysis is done on a single sample to ease notation, in the i.i.d. setting, simply sum over all the training samples to analyze the training set.

### 4.1 Empirical Monte-Carlo Convergence of Transformed Images

Given the close-form *average image* and *average loss*, we empirically measure how efficient is the MC estimation from Eq. (3).
We first propose in Fig. 4 a constructed $(64 \times 64)$ image for which we compute the expected loss (Eq. (12)) and the MC estimate (right-hand side of Eq. (3)). Surprisingly, we obtain that even for such a simple image and translation DA, between 1000 and 10000 samples are required to correctly estimate the MSE loss from the augmented samples. In a more practical scenario, one could rightfully argue that the combination of the DA samples from different images allows to obtain a better estimate with a smaller amount of augmentations per sample. Hence, we provide in Fig. 5 that experiment using a linear model on MNIST [45] with varying train set size. We observe that as the number of samples grows as the required number of augmentation per sample reduces. Nevertheless, even with thousands of samples, at least 50 augmentations per sample are required to provide an accurate estimate (reproduction with nonlinear models given in Fig. 7).
We now propose to specifically quantify the sensitivity of the MC estimate to DA sampling.

### 4.2 Loss Sensitivity Under Data-Augmentation Sampling in the Linear and Nonlinear Regime

Recalling Section 3.1, current DA training is performed on a MC estimate of the loss. The estimator's variance [46] is proportional to the variance of the quantity being estimated: $\mathbb{V}[(\mathcal{L} \circ f)(\mathcal{T}(\boldsymbol{x}))]$. We now characterize when, and why, would an MC estimator converge for a given model.

By leveraging the delta method [47, 48] i.e. a truncated Taylor expansion of the model and loss function mapping as $\mathcal{L} \circ f$, we have

$$\mathbb{V}[(\mathcal{L} \circ f)(\mathcal{T}(\boldsymbol{x}))] \approx \|\nabla(\mathcal{L} \circ f)(\mathbb{E}[\mathcal{T}(\boldsymbol{x})])\|_{\mathbb{V}[\mathcal{T}(\boldsymbol{x})]}^2, \quad (14)$$

with $\|\boldsymbol{u}\|_{\boldsymbol{A}}^2 \triangleq \boldsymbol{u}^T \boldsymbol{A} \boldsymbol{u}$. Noticing that $\mathbb{V}[\mathcal{T}(\boldsymbol{x})] = \mathbb{E}[\mathcal{T}(\boldsymbol{x})\mathcal{T}(\boldsymbol{x})^T] - \mathbb{E}[\mathcal{T}(\boldsymbol{x})]\mathbb{E}[\mathcal{T}(\boldsymbol{x})]^T$ and using the close-form moments of DST samples from Theorem 3.3, it is possible to write out Eq. (14) explicitly for model and DA specific analysis. We visualize $\mathbb{V}[\mathcal{T}(\boldsymbol{x})]$ in Fig. 3.

**Linear regression case.** To gain some insight into Eq. (14), let's first consider the linear regression case leading to $\mathbb{V}[(\mathcal{L} \circ f)(\mathcal{T}(\boldsymbol{x}))] \approx \|\boldsymbol{y} - \boldsymbol{W}\mathbb{E}[\mathcal{T}(\boldsymbol{x})] - \boldsymbol{b}\|_{\boldsymbol{W}\mathbb{V}[\mathcal{T}(\boldsymbol{x})]\boldsymbol{W}^T}^2$. Hence, the estimated loss variance depends (i) on the model predicting the correct output when observing the expected sample $\mathbb{E}[\mathcal{T}(\boldsymbol{x})]$, and (ii) on the smallest right singular vector of $\boldsymbol{W}$ to align with the largest eigenvectors of $\mathbb{V}[\mathcal{T}(\boldsymbol{x})]$, echoing our observation below Eq. (12).

---

[2]as commonly done, see e.g. Sec. A.2 from [16] for justification and approximation error analysis

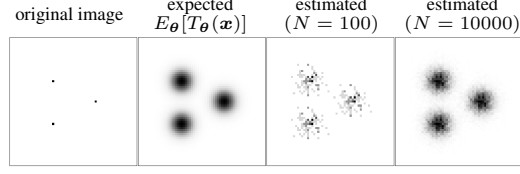

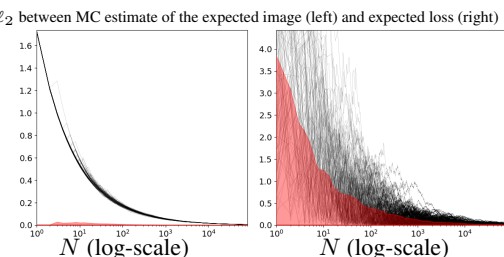

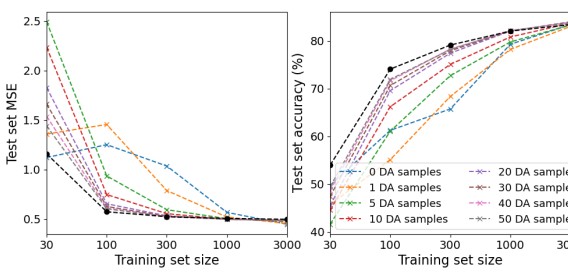

Figure 4: Analytical expected synthetic image under translation ($\mathcal{N}(0, 0.1) \otimes \mathcal{N}(0, 0.1)$) against its $N$-sample Monte-Carlo estimate (**top row**), $\ell_2$ distance between the true and estimated images (**bottom left**) and the true and estimated MSE loss with a random Gaussian $\boldsymbol{y}, \boldsymbol{W}$ (**bottom right**). In red is depicted the standard deviation of the independent Monte-Carlo runs. Clearly we observe that even on a simple $(64 \times 64)$ image and using the translation transformation, thousands of sampled images are necessary to provide an accurate estimate of the loss at hand. That is, sampling based data-augmentations are a rather inefficient medium to employ for injecting prior information into a model as this prior information will only emerge after tens of thousands of images have been sampled.

Figure 5: MNIST with linear model and translation DA with varying number of samples (**colored lines**) against the use of the expected loss (**black line**). We depict the MSE loss (**left**) and the accuracy (**right**) on the test set. At first, even a $50\times$ DA-based increase in dataset size fails to provide the performances reached by employing the analytical expected loss. As the training size increases, as the DA samples get redundant with the added samples closing the performance gap (the case of nonlinear model is provided in Fig. 7). See Tables 2 to 5 for additional table of results with standard deviation.

**General case.** Since the result from Eq. (14) holds in the general setting, we can generalize the above observation and formalize it into the following statement.

**Theorem 4.1.** *The variance of the loss's MC estimate (recall Eq. (3)) for an input $\boldsymbol{x}_n$ goes to $0$ if the loss gradient $\nabla \mathcal{L}(f(\mathbb{E}[\mathcal{T}(\boldsymbol{x})]))$ goes to $0$, or if the kernel of the model's Jacobian matrix $\boldsymbol{J}f(\mathbb{E}[\mathcal{T}(\boldsymbol{x})])$ aligns with the largest eigenvectors of the sample variance $\mathbb{V}[\mathcal{T}(\boldsymbol{x})]$.*

The proof of the above statement simply starts from Eq. (14) and applies the Cauchy-Schwarz inequality two times to obtain

$$\mathbb{V}[(\mathcal{L} \circ f)(\mathcal{T}(\boldsymbol{x}))] \leq \|\nabla \mathcal{L}(f(\mathbb{E}[\mathcal{T}(\boldsymbol{x})]))\|_2^4 \times \|\boldsymbol{J}f(\mathbb{E}[\mathcal{T}(\boldsymbol{x})])\boldsymbol{Q}(\boldsymbol{x})\Lambda^{\frac{1}{2}}(\boldsymbol{x})\|_F^4, \tag{15}$$

with decomposition $\mathbb{V}[\mathcal{T}(\boldsymbol{x})] = \boldsymbol{Q}(\boldsymbol{x})\Lambda(\boldsymbol{x})\boldsymbol{Q}(\boldsymbol{x})^T$. From Eq. (15), it is quite realistic to assume that the model's Jacobian does not collapse to $0$ (this would not minimize the training loss in general). Hence, the most realistic case of Theorem 4.1 concerns the alignment between the model's Jacobian matrix kernel space and the top eigenvectors of $\mathbb{V}[\mathcal{T}(\boldsymbol{x})]$. However, for models like ResNet DNs [49], the loss variance can only reduce to $0$ if the model minimizes the loss at the expected input since the Jacobian matrix is always full-rank. *We recall that the above relied on the delta method to approximate the intractable LHS of Eq. (14) (see footnote on Page 7).*

### 4.3 Explicit Loss Sensitivity Minimization Provably Recovers TangentProp

The expected loss (Eq. (3)) has been derived in the linear regression setting (Eq. (12)). But in a more general scenario, and as discussed below Theorem 3.4, this expectation might not be tractable and thus needs to be approximated e.g. based on a Taylor expansion of the loss and model. Alternatively to using the approximated expectation, one could employ the usual MC estimate of the expectation (right-hand side of Eq. (3)), and leverage the MC estimator variance obtained in Eq. (15) as a regularizer. We show here that both approaches are equivalent, and recover a popular regularizer known as TangentProp [50]. Although various extensions of TangentProp have been introduced [51, 52] no principled derivation of it has yet been proposed. Using the same argument as in Section 4.2, the expectation of a nonlinear transformation of a random variable ($\mathcal{T}(\boldsymbol{x}_n)$) can be

| | Train. size (N) | #DA=1 | #DA=2 | #DA=5 | #DA=10 | expectation (#DA=∞) |
|---|---|---|---|---|---|---|
| **rotation∈[-20,20] degrees, translations∈[-0.15,0.15]%, zoom∈[0.9, 1.1]%** — MNIST | N=100 | 10.39±1.82 | 11.70±2.02 | 10.60±1.34 | 24.30±1.92 | 33.44 |
| | N=1000 | 21.71±1.61 | 32.57±1.95 | 40.61±2.20 | 45.16±2.38 | 48.55 |
| | N=10000 | 47.82±2.20 | 51.98±1.33 | 54.97±0.92 | 56.29±0.66 | 57.16 |
| EMNIST | N=100 | 4.48±0.85 | 4.36±0.57 | 3.85±0.40 | 8.38±0.58 | 13.34 |
| | N=1000 | 8.45±0.53 | 14.79±0.60 | 22.51±0.63 | 27.49±0.60 | 32.18 |
| | N=10000 | 30.49±0.75 | 35.63±0.50 | 39.98±0.49 | 42.13±0.40 | 43.30 |
| FMNIST | N=100 | 11.18±3.31 | 9.49±1.40 | 10.65±1.66 | 23.10±2.17 | 35.31 |
| | N=1000 | 24.43±2.75 | 38.71±1.82 | 47.49±2.07 | 52.1±1.65 | 54.92 |
| | N=10000 | 52.60±2.57 | 54.9±1.70 | 56.84±1.42 | 57.29±1.44 | 57.63 |
| **rotation∈[-5,5] degrees, translations∈[-0.05,0.05]%, zoom∈[0.98, 1.02]%** — MNIST | N=100 | 12.72±1.95 | 12.42±2.02 | 12.45±1.47 | 45.82±1.53 | 55.83 |
| | N=1000 | 50.44±1.07 | 64.96±0.95 | 72.64±0.67 | 75.06±0.38 | 76.15 |
| | N=10000 | 78.54±0.22 | 79.76±0.24 | 80.49±0.20 | 80.85±0.17 | 80.86 |
| EMNIST | N=100 | 4.69±0.73 | 4.77±0.61 | 4.43±0.34 | 12.79±0.51 | 19.53 |
| | N=1000 | 16.74±0.47 | 31.10±0.29 | 40.59±0.39 | 44.45±0.34 | 46.53 |
| | N=10000 | 49.13±0.30 | 51.98±0.13 | 53.62±0.14 | 54.30±0.16 | 54.61 |
| FMNIST | N=100 | 11.51±2.12 | 12.43±1.99 | 11.59±1.40 | 37.08±1.23 | 55.22 |
| | N=1000 | 46.97±1.17 | 66.22±0.51 | 72.75±0.42 | 74.56±0.35 | 75.03 |
| | N=10000 | 75.2±0.33 | 75.89±0.27 | 76.49±0.32 | 76.63±0.18 | 76.71 |

Table 1: Reprise of Fig. 4 reporting test set classification performances averaged over 10 runs, with standard deviation corresponding to different DA realizations. For each specific run and configuration, all scenarios have access to the exact same training set, #DA represents the number of new samples introduced for each training sample, total training set size is thus N×(1+#DA). We observe that **in the large dataset size regime** ($N = 10000$) the gap between low number of DA samples and the expected regularizer is marginal which is expected as the introduced DA variations become redundant with the training set samples. This marginal gap can be slightly increased by employing stronger DA (compare top half and bottom half of the table). However, **in the small dataset size regime**, there is an important gain provided by employing the regularizer even when compared to observing many DA samples (#DA=10). This indicates a potentially crucial regime in which introducing the analytical regularizer helps. More surprisingly, this is also the case if part of DA policy is misaligned with the task at hand (e.g. rotation for FashionMNIST). Additional empirical results with more training set sizes ($N \in \{100, 200, 500, 1000, 3000, 10000, 20000\}$) and DA policies (one per table) are available in Tables 2 to 5 for MNIST, in Tables 6 to 9 for EMNIST, and in Tables 10 to 13 for FashionMNIST

approximated from the expectation of the Taylor expansion (detailed derivations in Appendix E) from which an upper-bound is obtained by applying the Cauchy-Schwarz inequality

$$\mathbb{E}[(\mathcal{L} \circ f)(\mathcal{T}(\boldsymbol{x}))] \leq (\mathcal{L} \circ f)(\mathbb{E}[\mathcal{T}(\boldsymbol{x})]) + \kappa(\boldsymbol{x}) \underbrace{\|\boldsymbol{J}f(\mathbb{E}[\mathcal{T}(\boldsymbol{x})])\boldsymbol{Q}(\boldsymbol{x})\Lambda(\boldsymbol{x})^{\frac{1}{2}}\|_F^2}_{\text{TangentProp regularization}},$$

with $\kappa(\boldsymbol{x}) \geq 0$ and the notations from Eq. (15). As a result, the TangentProp regularizer naturally appears when using a Taylor approximation of the expected loss, and it corresponds to adding an explicit loss variance regularization term (compare the TangentProp with Eq. (15)). We thus obtained from first principles that TangentProp emerges naturally when considering the second order Taylor approximation of the expected loss given a DA.

## 5 Conclusions and Limitations

In this paper, we proposed a novel set of mathematical tools (Sections 3.2 and 3.3) under which it is possible to study DA and to provably answer some of the open questions around the efficiency and impact of DA to train a model. We first obtained the explicit regularizer produced by different DAs in Section 3.4. This led to the following observation: the kernel space of the $\boldsymbol{W}$ matrix is pushed to align with the largest eigenvectors of the sample covariance matrix. This was then studied in a more general setting in Section 4.2 for nonlinear models when characterizing the loss variance under DA sampling. We also observed that Monte-Carlo sampling of transformed images is highly inefficient (Section 4.1), even if similar training samples combine their underlying information within a dataset. Lastly, those derivations led us to provably derive a known regularizer —TangentProp— as being the natural minimizer of a model's loss variance. The main limitation of the current analysis concerns nonlinear models for which we rely on a third-order Taylor expansion.

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
