# Appendix:
# A Data-Augmentation Is Worth A Thousand Samples

The appendix provides additional supporting materials such as figures, proofs, and additional discussions.

## A Experiment Details

For the experiments of Fig. 5 we employed SGD optimizer with initial learning rate of $0.1$, nesterov momentum of $0.9$, training from $2000$ steps and with a learning rate multiplier of $0.3$ are step $700$ and $1400$. The number of training samples and the number of DA samples is specified in the figure. The process is simple. For a given training set set N, and number of DA samples D, we generate a N×D large training set where the given DA policy is applied $D$ times on each training sample. For the case of Fig. 4 the analytical least square solution is employed (with DA samples or with expected regularizer).

| dataset size (ds) \ DA samples | ds × 1 | ds ×2 | ds ×5 | ds ×10 | expectation |
|---|---|---|---|---|---|
| ds=100 | 10.39±1.82 | 11.7±2.02 | 10.6 ±1.34 | 24.3±1.92 | 33.44 |
| ds=200 | 11.44±1.97 | 11.84±1.30 | 23.32±1.74 | 33.32±1.64 | 38.57 |
| ds=500 | 11.21±1.45 | 22.95±1.28 | 35.59±2.12 | 41.39±1.94 | 44.31 |
| ds=1000 | 21.71±1.61 | 32.57±1.95 | 40.61±2.20 | 45.16±2.38 | 48.55 |
| ds=3000 | 37.25±2.47 | 43.42±2.71 | 48.88±1.98 | 51.82±1.84 | 53.36 |
| ds=10000 | 47.82±2.20 | 51.98±1.33 | 54.97±0.92 | 56.29±0.66 | 57.16 |
| ds=20000 | 51.65±1.04 | 54.83±0.87 | 57.15±0.78 | 57.94±0.60 | 58.27 |

Table 2: **MNIST:** Reprise of Fig. 4 but now with augmentation range of $[-20, 20]$ for rotation angle, $[-0.15\%, 0.15\%]$ for vertical and horizontal translation, and of $[0.9, 1.1]$ for zoom range. Standard deviation corresponds to different DA realization over 10 successive runs. The only variation between runs is the realization of the DAs (except for the expectation scenario in which the explicit regularizer is employed and thus no randomness is present).

| dataset size (ds) \ DA samples | ds × 1 | ds ×2 | ds ×5 | ds ×10 | expectation |
|---|---|---|---|---|---|
| ds=100 | 11.38±1.79 | 12.48±1.12 | 12.62±2.08 | 46.70±0.34 | 53.77 |
| ds=200 | 11.98±1.10 | 12.86±1.60 | 51.85±1.48 | 60.66±0.88 | 63.27 |
| ds=500 | 12.78±2.43 | 56.21±1.45 | 68.75±0.66 | 71.27±0.43 | 72.22 |
| ds=5000 | 55.76±1.01 | 68.23±0.48 | 73.63±0.48 | 75.00±0.43 | 75.70 |
| ds=3000 | 74.19±0.45 | 76.86±0.27 | 78.34±0.26 | 78.84±0.29 | 79.02 |
| ds=10000 | 78.72±0.19 | 79.48±0.21 | 79.76±0.12 | 80.02±0.11 | 80.10 |
| ds=20000 | 80.25±0.21 | 80.60±0.18 | 80.84±0.15 | 80.95±0.11 | 80.96 |

Table 3: **MNIST:** Reprise of Table 2 but now with augmentation range of $[-20, 20]$ for rotation angle. Standard deviation corresponds to different DA realization, over 10 successive runs.

## B Additional Figures

We first propose a visual depiction of coordinate space image transformations in Fig. 6. We employ here the case of a zoom transformation on a randomly generated image for illustration purposes. It can be seen that the transformation acts by altering the position of the image coordinates, and then produce the estimated value of the image at those new coordinates through some interpolation schemes. The most simple strategy is to employ a nearest-neighbor estimate i.e. given the new coordinate, predict the value of the new image at that position to be the same as the pixel value of

| dataset size (ds) \ DA samples | ds × 1 | ds ×2 | ds ×5 | ds ×10 | expectation |
|---|---|---|---|---|---|
| ds=100 | 10.19±2.43 | 10.90±2.03 | 11.08±1.42 | 14.41±1.13 | 17.85 |
| ds=200 | 10.50±1.71 | 10.49±1.31 | 14.81±1.83 | 18.07±2.26 | 19.47 |
| ds=500 | 10.51±1.47 | 15.48±1.87 | 18.98±2.06 | 20.91±2.06 | 21.77 |
| ds=1000 | 15.36±2.26 | 17.55±2.43 | 20.28±1.28 | 22.45±1.82 | 23.67 |
| ds=3000 | 19.04±2.49 | 21.00 ±2.63 | 23.96±1.60 | 24.34±1.51 | 24.49 |
| ds=10000 | 22.84±2.26 | 25.35±0.87 | 25.74±1.02 | 26.10±1.26 | 26.31 |
| ds=20000 | 23.92±1.44 | 25.21±1.34 | 25.68±1.22 | 25.52±0.78 | 25.46 |

Table 4: **MNIST:** Reprise of Table 2 but now with augmentation range of $[-40, 40]$ for rotation angle, $[-0.2\%, 0.2\%]$ for vertical and horizontal translation, and of $[0.8, 1.2]$ for zoom range. Standard deviation corresponds to different DA realization, over 10 successive runs.

| dataset size (ds) \ DA samples | ds × 1 | ds ×2 | ds ×5 | ds ×10 | expectation |
|---|---|---|---|---|---|
| ds=100 | 12.72±1.95 | 12.42±2.02 | 12.45±1.47 | 45.82±1.53 | 55.83 |
| ds=200 | 11.94±1.85 | 11.93±1.45 | 49.11±0.85 | 60.17±0.87 | 63.77 |
| ds=500 | 11.30±1.48 | 52.07±1.11 | 66.81±0.76 | 70.69±0.47 | 72.17 |
| ds=1000 | 50.44±1.07 | 64.96±0.95 | 72.64±0.67 | 75.06±0.38 | 76.15 |
| ds=3000 | 71.54±0.66 | 76.03±0.31 | 78.29±0.20 | 79.01±0.32 | 79.35 |
| ds=10000 | 78.54±0.22 | 79.76±0.24 | 80.49±0.20 | 80.85±0.17 | 80.86 |
| ds=20000 | 80.42±0.37 | 81.13±0.25 | 81.44±0.18 | 81.56±0.12 | 81.57 |

Table 5: **MNIST:** Reprise of Table 2 but now with augmentation range of $[-5, 5]$ for rotation angle, $[-0.05\%, 0.05\%]$ for vertical and horizontal translation, and of $[0.98, 1.02]$ for zoom range. Standard deviation corresponds to different DA realization, over 10 successive runs.

| dataset size (ds) \ DA samples | ds × 1 | ds × 2 | ds × 5 | ds × 10 | ds × 20 |
|---|---|---|---|---|---|
| ds=100 | 4.48±0.85 | 4.36±0.57 | 3.85±0.40 | 8.38±0.58 | 13.34 |
| ds=200 | 3.63±0.34 | 4.21±0.46 | 8.95±0.40 | 14.60±0.49 | 18.39 |
| ds=500 | 4.10±0.47 | 8.15±0.60 | 16.18±0.75 | 20.28±0.73 | 24.05 |
| ds=1000 | 8.45±0.53 | 14.79±0.6 | 22.51±0.63 | 27.49±0.60 | 32.18 |
| ds=3000 | 19.53±0.89 | 25.72±0.64 | 32.84±0.60 | 36.81±0.88 | 39.69 |
| ds=10000 | 30.49±0.75 | 35.63±0.50 | 39.98±0.49 | 42.13±0.40 | 43.30 |
| ds=20000 | 35.63±0.47 | 39.27±0.40 | 42.25±0.41 | 43.63±0.50 | 44.42 |

Table 6: **EMNIST:** Reprise of Table 2 but now with on EMNIST and now with augmentation range of $[-20, 20]$ for rotation angle, $[-0.15\%, 0.15\%]$ for vertical and horizontal translation, and of $[0.9, 1.1]$ for zoom range. Standard deviation corresponds to different DA realization over 10 successive runs. The only variation between runs is the realization of the DAs (except for the expectation scenario in which the explicit regularizer is employed and thus no randomness is present).

| dataset size (ds) \ DA samples | ds × 1 | ds × 2 | ds × 5 | ds × 10 | ds × 20 |
|---|---|---|---|---|---|
| ds=100 | 4.47±0.55 | 4.74±0.62 | 4.55±0.38 | 13.74±0.4 | 17.94 |
| ds=200 | 4.45±0.44 | 4.51±0.4 | 15.13±0.33 | 22.2±0.48 | 24.56 |
| ds=500 | 4.48±0.39 | 17.46±0.46 | 29.92±0.31 | 33.39±0.35 | 35.05 |
| ds=1000 | 18.88±0.42 | 31.97±0.62 | 38.95±0.33 | 41.4±0.25 | 42.69 |
| ds=3000 | 39.56±0.5 | 45.1±0.27 | 48.91±0.24 | 50.29±0.16 | 51.02 |
| ds=10000 | 50.05±0.18 | 52.29±0.19 | 53.73±0.2 | 54.32±0.16 | 54.61 |
| ds=20000 | 52.95±0.22 | 54.36±0.19 | 55.17±0.17 | 55.43±0.15 | 55.6 |

Table 7: **EMNIST:** Reprise of Table 6 but now with augmentation range of $[-20, 20]$ for rotation angle. Standard deviation corresponds to different DA realization, over 10 successive runs.

the closest coordinate. As the coordinate transformation is composed with the image interpolation scheme, it is intricate to obtain any analytical quantity such as the expected image, motivating our proposed pixel-space transformation.

| dataset size (ds) \ DA samples | ds × 1 | ds × 2 | ds × 5 | ds × 10 | ds × 20 |
|---|---|---|---|---|---|
| ds=100 | 4.24±0.50 | 4.33±0.78 | 3.75±0.46 | 5.84±0.43 | 7.74 |
| ds=200 | 4.06±0.48 | 4.10±0.47 | 5.98±0.73 | 8.51±0.68 | 10.25 |
| ds=500 | 3.94±0.37 | 5.80±0.61 | 9.05±0.63 | 11.02±0.59 | 12.91 |
| ds=1000 | 5.96±0.56 | 8.60±0.50 | 11.48±0.77 | 13.75±0.82 | 16.47 |
| ds=3000 | 9.89±0.87 | 12.63±0.80 | 16.86±0.85 | 19.87±0.58 | 23.04 |
| ds=10000 | 15.18±1.12 | 18.56±1.16 | 22.87±1.06 | 25.38±0.78 | 26.89 |
| ds=20000 | 18.39±0.79 | 21.55±0.79 | 25.31±0.73 | 27.0±0.81 | 28.02 |

Table 8: **EMNIST:** Reprise of Table 6 but now with augmentation range of $[-40, 40]$ for rotation angle, $[-0.2\%, 0.2\%]$ for vertical and horizontal translation, and of $[0.8, 1.2]$ for zoom range. Standard deviation corresponds to different DA realization, over 10 successive runs.

| dataset size (ds) \ DA samples | ds × 1 | ds × 2 | ds × 5 | ds × 10 | ds × 20 |
|---|---|---|---|---|---|
| ds=100 | 4.69±0.73 | 4.77±0.61 | 4.43±0.34 | 12.79±0.51 | 19.53 |
| ds=200 | 4.84±0.70 | 4.13±0.36 | 14.32±0.51 | 23.72±0.30 | 27.61 |
| ds=500 | 4.40±0.34 | 15.68±0.52 | 30.68±0.32 | 35.74±0.17 | 38.10 |
| ds=1000 | 16.74±0.47 | 31.10±0.29 | 40.59±0.39 | 44.45±0.34 | 46.53 |
| ds=3000 | 37.81±0.34 | 44.53±0.24 | 49.43±0.18 | 51.30±0.20 | 52.29 |
| ds=10000 | 49.13±0.30 | 51.98±0.13 | 53.62±0.14 | 54.30±0.16 | 54.61 |
| ds=20000 | 52.40±0.22 | 54.10±0.20 | 55.11±0.14 | 55.46±0.14 | 55.65 |

Table 9: **EMNIST:** Reprise of Table 6 but now with augmentation range of $[-5, 5]$ for rotation angle, $[-0.05\%, 0.05\%]$ for vertical and horizontal translation, and of $[0.98, 1.02]$ for zoom range. Standard deviation corresponds to different DA realization, over 10 successive runs.

| dataset size (ds) \ DA samples | ds × 1 | ds × 2 | ds × 5 | ds × 10 | ds × 20 |
|---|---|---|---|---|---|
| ds=100 | 11.18±3.31 | 9.49±1.40 | 10.65±1.66 | 23.10±2.17 | 35.31 |
| ds=200 | 10.64±1.90 | 11.07±1.99 | 23.93±1.87 | 37.72±2.37 | 44.90 |
| ds=500 | 10.27±0.97 | 24.44±1.65 | 40.69±1.97 | 46.55±1.73 | 51.48 |
| ds=1000 | 24.43±2.75 | 38.71±1.82 | 47.49±2.07 | 52.1±1.65 | 54.92 |
| ds=3000 | 44.23±1.64 | 49.19±1.35 | 53.88±1.52 | 55.32±1.74 | 56.12 |
| ds=10000 | 52.60±2.57 | 54.9±1.70 | 56.84±1.42 | 57.29±1.44 | 57.63 |
| ds=20000 | 55.11±1.83 | 56.62±1.75 | 57.21±1.41 | 57.55±1.22 | 57.59 |

Table 10: **F-MNIST:** Reprise of Table 2 but now with on F-MNIST and now with augmentation range of $[-20, 20]$ for rotation angle, $[-0.15\%, 0.15\%]$ for vertical and horizontal translation, and of $[0.9, 1.1]$ for zoom range. Standard deviation corresponds to different DA realization over 10 successive runs. The only variation between runs is the realization of the DAs (except for the expectation scenario in which the explicit regularizer is employed and thus no randomness is present).

| dataset size (ds) \ DA samples | ds × 1 | ds × 2 | ds × 5 | ds × 10 | ds × 20 |
|---|---|---|---|---|---|
| ds=100 | 10.96±1.69 | 10.79±1.68 | 11.55±1.22 | 40.44±1.18 | 49.71 |
| ds=200 | 12.25±1.84 | 13.03±1.52 | 44.98±1.07 | 57.95±0.55 | 61.22 |
| ds=500 | 11.84±1.66 | 46.71±1.75 | 64.78±0.68 | 67.77±0.44 | 69.22 |
| ds=1000 | 48.10±1.10 | 64.77±0.75 | 70.38±0.32 | 72.03±0.35 | 72.82 |
| ds=3000 | 69.61±0.52 | 72.51±0.53 | 73.94±0.26 | 74.41±0.23 | 74.59 |
| ds=10000 | 74.16±0.22 | 74.81±0.24 | 75.04±0.19 | 75.10±0.08 | 75.11 |
| ds=20000 | 74.75±0.24 | 75.24±0.24 | 75.46±0.09 | 75.41±0.10 | 75.43 |

Table 11: **F-MNIST:** Reprise of Table 10 but now with augmentation range of $[-20, 20]$ for rotation angle. Standard deviation corresponds to different DA realization, over 10 successive runs.

## C  Additional Data-Augmentations

The case of **Random Crop** can be studied with the already derived DAs through composition. In fact, a random crop can be seen as composing a translation, possibly a shearing (if the aspect ratio of the crop is not preserved) and a zooming operation. Since it is common to have random and

| dataset size (ds) | ds × 1 | ds × 2 | ds × 5 | ds × 10 | ds × 20 |
|---|---|---|---|---|---|
| ds=100 | 9.88±2.67 | 9.82±1.70 | 10.15±0.98 | 16.4±1.62 | 23.95 |
| ds=200 | 10.24±2.53 | 10.55±2.33 | 15.21±2.93 | 23.62±1.99 | 27.95 |
| ds=500 | 10.21±1.75 | 17.35±1.90 | 25.09±1.46 | 29.85±1.57 | 33.39 |
| ds=1000 | 17.41±1.98 | 25.8±2.22 | 32.05±1.20 | 37.05±1.03 | 39.77 |
| ds=3000 | 27.50±1.84 | 31.49±1.63 | 36.34±1.48 | 38.36±1.35 | 39.90 |
| ds=10000 | 34.09±2.10 | 37.92±2.59 | 41.36±2.33 | 42.5±1.40 | 42.64 |
| ds=20000 | 37.72±1.68 | 40.05±1.82 | 42.07±1.79 | 42.94±1.33 | 43.61 |

Table 12: **F-MNIST:** Reprise of Table 10 but now with augmentation range of $[-40, 40]$ for rotation angle, $[-0.2\%, 0.2\%]$ for vertical and horizontal translation, and of $[0.8, 1.2]$ for zoom range. Standard deviation corresponds to different DA realization, over 10 successive runs.

| dataset size (ds) | ds × 1 | ds × 2 | ds × 5 | ds × 10 | ds × 20 |
|---|---|---|---|---|---|
| ds=100 | 11.51±2.12 | 12.43±1.99 | 11.59±1.40 | 37.08±1.23 | 55.22 |
| ds=200 | 12.40±1.27 | 11.80±1.98 | 43.0±1.09 | 61.70±0.52 | 66.27 |
| ds=500 | 11.57±1.33 | 45.08±1.43 | 67.09±0.66 | 70.92±0.51 | 72.36 |
| ds=1000 | 46.97±1.17 | 66.22±0.51 | 72.75±0.42 | 74.56±0.35 | 75.03 |
| ds=3000 | 70.63±0.46 | 74.04±0.64 | 75.53±0.50 | 75.78±0.23 | 75.88 |
| ds=10000 | 75.2±0.33 | 75.89±0.27 | 76.49±0.32 | 76.63±0.18 | 76.71 |
| ds=20000 | 76.16±0.23 | 76.55±0.22 | 76.79±0.16 | 76.76±0.12 | 76.82 |

Table 13: **F-MNIST:** Reprise of Table 10 but now with augmentation range of $[-5, 5]$ for rotation angle, $[-0.05\%, 0.05\%]$ for vertical and horizontal translation, and of $[0.98, 1.02]$ for zoom range. Standard deviation corresponds to different DA realization, over 10 successive runs.

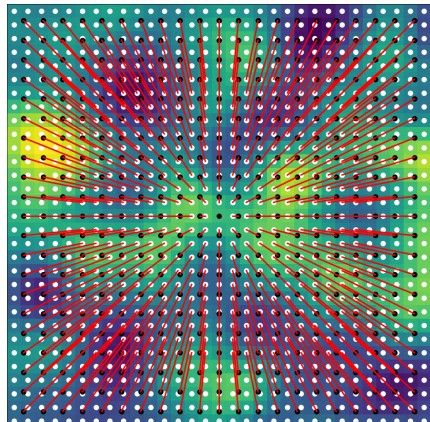

Figure 6: Depiction of the coordinate space transformation applied on a random image (white noise convolved with a low-pass filter. In this setting, one starts from the original image coordinates (white dots) and uses the mapping $t$ (recall Eq. (1)) to know what coordinate $t(u, v)$ is associated to the target image coordinate $(u, v)$. In this example of a zoom transformation, the $t$ mapping will scale down those coordinates leading to a displacement (red lines) producing the black dots from the white ones. Once the new coordinates $(t(u, v))$ are known, the actual value is obtained through some flavors of interpolation between the white dots that are the closest to each black dot. For example, using the pixel value of the closest white dot corresponds to a nearest neighbor interpolation. Commonly, the 4 closest pixels are used with a bilinear weighting.

independent parameters for those, one can simply obtain the matrix $M$ of random crop by matrix-multiplying the matrices of those transformations. To see that, notice that due to independence of those transformations, the expectation can be taken separately for each transformation.

The case of **CutOut, CutMix or MixUp** offer interesting avenues to our work. We omit those augmentations for this study due to the fact that such DAs involve an additional transformation of the target variable that is not independent from the transformation applied on the image. That is, even in

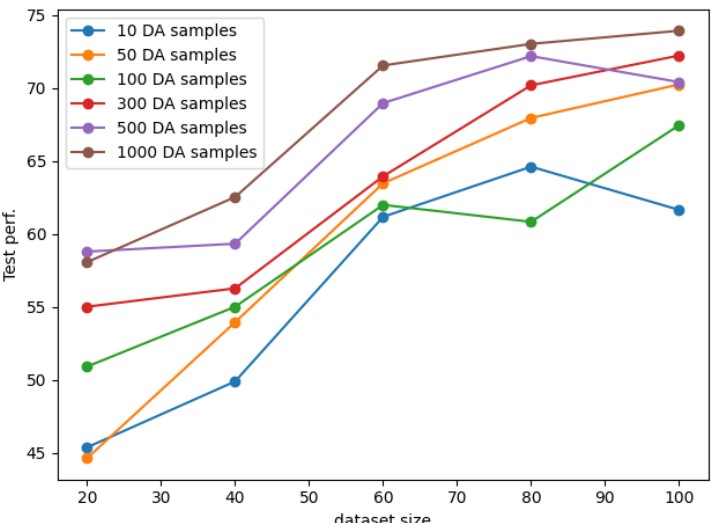

Figure 7: Reprise of Fig. 5 using a nonlinear network (LeNet5 with ReLU) with varying DA samples and dataset size on MNIST, reporting the same type of convergence than in the linear regime i.e. hundreds of DA samples are needed for the model to incorporate its symmetries.

the linear regime, unrolling the MSE introduces the inner product between the linearly transformed DA input, and the target variable. Taking the expectation of this inner product would require the joint density that encodes both the transformation of the input and the output. Hence, all the proposed results would need to be derived that way. We thus leave such analysis for future work.

The case of **Grayscale** transformation. This augmentation is much simpler to deal with that other spatial DAs for a simple reason: it applies its transformation pixel-wise by combining the RGB channels of an image. There are different ways to combine those channels to produce the grayscale image, the simpler being to average them. Consider this mapping to be the $g : \mathbb{R}^3 \mapsto \mathbb{R}^3$ mapping that is applied at each pixel. Consider the DA to have the probability to apply grayscale with probability $\theta$ (Bernoulli variable simple this DA is either applied or not). Now, using the same formulation as for the affine case, one can directly obtain that the expected image is a combex combination of the original image $I$ and its grayscale counterpart $G$ as in $G\theta + (1 - \theta)I$. The exact same analysis can be carried out for the case of **Random Contrast** as used in color jitter which consists in mixing the natural image $I$ with its grayscale counterpart $G$ via a uniformly sampled coefficient. In that case and again using the same derivation we obtain that the expected image would be $Ic + G(1 - c)$ with $c$ the mean of the employed uniform distribution. And so the **color jitter** case simply combines multiple such transforms independently one after the other, meaning that the expectation can be taken one at a time using the above procedure to obtain the expected color jitter DA. The case of **random Gaussian blur** can also be obtained as in the grayscale case to obtain the convex combination of the original image and its blurred counterpart with coefficient being given by the mean of the Bernoulli random variable that decides if the DA is applied.

# D   Expected Mean-Squared Error Derivation

In this section, we provide through direct derivations the expression of the expected mean squared error, under a given data augmentation distribution, as a function of the image first two moments. This result is crucial as it demonstrates that knowledge of those two moments is sufficient to obtain a closed form solution of the expected loss. As far as we are aware, this is a new result enabled by the proposed pixel-space transformation.

Using the cylic property of the Trace operator and the fact that $\mathbb{V}[\mathcal{T}(\boldsymbol{x})] = \mathbb{E}[\mathcal{T}(\boldsymbol{x})\mathcal{T}(\boldsymbol{x})^T] - \mathbb{E}[\mathcal{T}(\boldsymbol{x})]\mathbb{E}[\mathcal{T}(\boldsymbol{x})]^T$ we can express the expected MSE loss as the MSE loss between the target and

the expected input plus a regularization term as follows

$$
\begin{aligned}
\mathcal{L}(\boldsymbol{W}, \boldsymbol{b}) =& \sum_{n=1}^{N} \mathbb{E}_{\theta \sim \Theta} \left[ \| \boldsymbol{y}_n - \boldsymbol{W} \mathcal{T}_\theta(\boldsymbol{x}_n) - \boldsymbol{b} \|_2^2 \right] \\
=& \sum_{n=1}^{N} \| \boldsymbol{y}_n - \boldsymbol{b} \|_2^2 - 2 \langle \boldsymbol{W}^T (\boldsymbol{y}_n - \boldsymbol{b}), \mathbb{E}_{\theta \sim \Theta} \left[ \mathcal{T}_\theta(\boldsymbol{x}_n) \right] \rangle + \mathbb{E}_{\theta \sim \Theta} \left[ \mathcal{T}_\theta(\boldsymbol{x}_n)^T \boldsymbol{W}^T \boldsymbol{W} \mathcal{T}_\theta(\boldsymbol{x}_n) \right] \\
=& \sum_{n=1}^{N} \| \boldsymbol{y}_n - \boldsymbol{b} \|_2^2 - 2 \langle \boldsymbol{W}^T (\boldsymbol{y}_n - \boldsymbol{b}), \mathbb{E}_{\theta \sim \Theta} \left[ \mathcal{T}_\theta(\boldsymbol{x}_n) \right] \rangle \\
& + \mathbb{E}_{\theta \sim \Theta} \left[ \mathrm{Tr} \left( \boldsymbol{W}^T \boldsymbol{W} \mathcal{T}_\theta(\boldsymbol{x}_n) \mathcal{T}_\theta(\boldsymbol{x}_n)^T \right) \right] \quad \text{(cyclic prop.)} \\
=& \sum_{n=1}^{N} \| \boldsymbol{y}_n - \boldsymbol{b} \|_2^2 - 2 \langle \boldsymbol{W}^T (\boldsymbol{y}_n - \boldsymbol{b}), \mathbb{E}_{\theta \sim \Theta} \left[ \mathcal{T}_\theta(\boldsymbol{x}_n) \right] \rangle \\
& + \mathrm{Tr} \left( \boldsymbol{W}^T \boldsymbol{W} \mathbb{E}_{\theta \sim \Theta} \left[ \mathcal{T}_\theta(\boldsymbol{x}_n) \mathcal{T}_\theta(\boldsymbol{x}_n)^T \right] \right) \quad \text{(linear prop.)} \\
=& \sum_{n=1}^{N} \| \boldsymbol{y}_n - \boldsymbol{b} \|_2^2 + \langle \boldsymbol{W}^T (\boldsymbol{y}_n - \boldsymbol{b}), \mathbb{E}_{\theta \sim \Theta} \left[ \mathcal{T}_\theta(\boldsymbol{x}_n) \right] \rangle \\
& + \mathrm{Tr} \left( \boldsymbol{W}^T \boldsymbol{W} \left( \mathbb{E}_{\theta \sim \Theta} \left[ \mathcal{T}_\theta(\boldsymbol{x}_n) \mathcal{T}_\theta(\boldsymbol{x}_n)^T \right] - \mathbb{E}_{\theta \sim \Theta} \left[ \mathcal{T}_\theta(\boldsymbol{x}_n) \right] \mathbb{E}_{\theta \sim \Theta} \left[ \mathcal{T}_\theta(\boldsymbol{x}_n) \right]^T \right) \right) \\
& + \mathrm{Tr} \left( \boldsymbol{W}^T \boldsymbol{W} \left( \mathbb{E}_{\theta \sim \Theta} \left[ \mathcal{T}_\theta(\boldsymbol{x}_n) \right] \mathbb{E}_{\theta \sim \Theta} \left[ \mathcal{T}_\theta(\boldsymbol{x}_n) \right]^T \right) \right) \\
=& \sum_{n=1}^{N} \| \boldsymbol{y}_n - \boldsymbol{b} \|_2^2 + \langle \boldsymbol{W}^T (\boldsymbol{y}_n - \boldsymbol{b}), \mathbb{E}_{\theta \sim \Theta} \left[ \mathcal{T}_\theta(\boldsymbol{x}_n) \right] \rangle \\
& + \mathrm{Tr} \left( \boldsymbol{W}^T \boldsymbol{W} \mathbb{E}_{\theta \sim \Theta} \left[ \mathcal{T}_\theta(\boldsymbol{x}_n) \right] \mathbb{E}_{\theta \sim \Theta} \left[ \mathcal{T}_\theta(\boldsymbol{x}_n) \right]^T \right) \\
& + \mathrm{Tr} \left( \boldsymbol{W}^T \boldsymbol{W} \mathbb{V}_{\theta \sim \Theta} \left[ \mathcal{T}_\theta(\boldsymbol{x}_n) \right] \right) \\
=& \sum_{n=1}^{N} \| \boldsymbol{y}_n - \boldsymbol{W} \mathbb{E}_{\theta \sim \Theta} \left[ \mathcal{T}_\theta(\boldsymbol{x}_n) \right] - \boldsymbol{b} \|_2^2 + \mathrm{Tr} \left( \boldsymbol{W}^T \boldsymbol{W} \mathbb{V}_{\theta \sim \Theta} \left[ \mathcal{T}_\theta(\boldsymbol{x}_n) \right] \right),
\end{aligned}
$$

concluding our derivations. Notice how the regularization term acts upon the matrix $\boldsymbol{W}$ through the variance of the image under the specified transformation.

# E   Taylor Approximation

In this section we now describe how to exploit a second order Taylor approximation of any loss and/or transformation of the transformed image as a way to obtain an approximated expected loss/output. Without loss of generality we consider here this mapping to be $(\mathcal{L} \circ f)$, i.e. a nonlinear mapping $f$ and a loss function $\mathcal{L}$. The same result applies regardless of those mappings. The approximation will holds mostly for small transformation due to the quadratic approximation of the mapping.

Let's first obtain the second order Taylor expansion of the nonlinear mapping at $\mathbb{E}[\mathcal{T}(\boldsymbol{x})]$ as

$$
\begin{aligned}
(\mathcal{L} \circ f)(\mathcal{T}(\boldsymbol{x})) \approx& (\mathcal{L} \circ f)(\mathbb{E}[\mathcal{T}(\boldsymbol{x})]) + (\mathcal{T}(\boldsymbol{x}_n) - \mathbb{E}[\mathcal{T}(\boldsymbol{x})])^T \nabla (\mathcal{L} \circ f)(\mathbb{E}[\mathcal{T}(\boldsymbol{x})]) \\
& + \frac{1}{2} (\mathcal{T}(\boldsymbol{x}_n) - \mathbb{E}[\mathcal{T}(\boldsymbol{x})])^T H(\mathcal{L} \circ f)(\mathbb{E}[\mathcal{T}(\boldsymbol{x})])(\mathcal{T}(\boldsymbol{x}_n) - \mathbb{E}[\mathcal{T}(\boldsymbol{x})])
\end{aligned}
$$

notice that we are required to at least take the second order Taylor expansion as the first order will vanish as soon as we will take the expectation of that approximation. In fact, taking the expectation

of the above, we obtain (using the cyclic property of the Trace operator)

$$
\begin{aligned}
\mathbb{E}[(\mathcal{L} \circ f)(\mathcal{T}(\boldsymbol{x}))] \approx & (\mathcal{L} \circ f)(\mathbb{E}[\mathcal{T}(\boldsymbol{x})]) + \frac{1}{2}\operatorname{Tr}\left(H(\mathcal{L} \circ f)(\mathbb{E}[\mathcal{T}(\boldsymbol{x})])\mathbb{V}(\mathcal{T}(\boldsymbol{x}))\right) \\
= & (\mathcal{L} \circ f)(\mathbb{E}[\mathcal{T}(\boldsymbol{x})]) \\
& + \frac{1}{2}\operatorname{Tr}\left(\boldsymbol{J}f_\gamma(\mathbb{E}[\mathcal{T}(\boldsymbol{x})]^T H\mathcal{L}(f_\gamma(\mathbb{E}[\mathcal{T}(\boldsymbol{x})])\boldsymbol{J}f_\gamma(\mathbb{E}[\mathcal{T}(\boldsymbol{x})]\mathbb{V}(\mathcal{T}(\boldsymbol{x})))\right) \\
= & (\mathcal{L} \circ f)(\mathbb{E}[\mathcal{T}(\boldsymbol{x})]) + \frac{1}{2}\|\boldsymbol{U}^{\frac{1}{2}}(\boldsymbol{x})\boldsymbol{V}(\boldsymbol{x})^T\boldsymbol{J}f_\gamma(\mathbb{E}[\mathcal{T}(\boldsymbol{x})])\boldsymbol{Q}(\boldsymbol{x})\Lambda(\boldsymbol{x})^{\frac{1}{2}}\|_F^2,
\end{aligned}
$$

where $H$ represents the Hessian. The above concludes our derivation that led to a generalization of the linear case. In fact, notice that in the linear with mean squared error the second order Taylor approximation is exact and the above is exactly the same as the one of the previous section.

## F Proof of Thm. 3.3

This section provides the derivation of the first two moments of images under specific image transformations. Note that the actual distribution is abstracted away as simple $p$. In fact, those results do not depend on the specific form of $p$, rather, they depend on the type of transformation being applied e.g. rotation, translation or zoom. We thus propose to derive them, following the same recipe one will be able to obtain the analytical form of the first two moments for any desired transformation. One fact that we will heavily leverage is the fact that integrating a functional and a Dirac function can be expressed as evaluating that function as the position of the Dirac (recall Proposition 3.2).

Throughout this proof, we will denote by $T(u, v; \theta)$ the value of the transformed image at spatial position $(u, v)$, hence $\mathbb{E}_{\theta \sim \Theta}[T(u, v; \theta)]$ is the expected value of the transformed image at pixel position $(u, v)$. And $\mathbb{E}_{\theta \sim \Theta}[T(u, v; \theta)T(u', v'; \theta)]$ is the second order (uncentered) moment representing the interplay between pixel positions $(u, v)$ and $(u'v')$ of the transformed image. This second order moment and the first order moment can be used to obtained the variance/covariance of the transformed image.

**Vertical and Horizontal translation:** The case of vertical and horizontal translations is taken care of jointly, for vertical-only or horizontal-only transformations, simply use a distribution that is a Dirac (at 0) for the transformation that is not needed. We thus obtain in general given a 2-dimensional density $p$ as

$$
\begin{aligned}
\mathbb{E}_{\theta \sim \Theta}[T(u, v; \theta)] &= \int_\theta p(\theta) \int I(x, y)h_\theta(u, v, x, y)dxdyd\theta \\
&= \int_{-\infty}^{\infty} p(\theta_1, \theta_2) \int I(x, y)\delta(u = x + \theta_1, v = y + \theta_2)dxdyd\theta \\
&= \int I(x, y) \int_{-\infty}^{\infty} p(\theta_1, \theta_2)\delta(u = x + \theta_1, v = y + \theta_2)d\theta dxdy \\
&= \int I(x, y)p(u - x, v - y)dxdy
\end{aligned}
$$

from this we also directly obtain that the expected image can be expressed as a 2-dimensional convolution between the original image and the density being employed for the translation as in

$\mathbb{E}_{\theta \sim \Theta}[T(.,.;\theta)] = I \star p$. We now derive the second order moments

$$\mathbb{E}_{\theta \sim \Theta}[T(u,v;\theta)T(u',v';\theta)] = \int I(x,y)I(x',y')\int_{-\infty}^{\infty} p(\theta_1,\theta_2)\delta(u = x + \theta_1, v = y + \theta_2)$$

$$\times \ \delta(u' = x' + \theta_1, v' = y' + \theta_2)d\theta_1 d\theta_2 dx dy dx' dy'$$

$$= \int I(x,y)I(x',y')p(u-x,v-y)\delta(v-y-(v'-y'))\delta(u-x-(u'-x'))dxdx'dydy'$$

$$= \int I(x'-u'+u,y'-v'+v)I(x',y')p(u-x'+u'-u,v-(y'-v'+v))dx'dy'$$

$$= \int I(x'-u'+u,y'-v'+v)I(x',y')p(-x'+u',-y'+v')dx'dy'$$

$$= \int I(a+u,b+v)I(a+u',b+v')p(-a,-b)dadb,$$

as a result the second order moment of the image at $(u,v)$ and $(u',v')$ is simply the inner product between the image translated to $(u,v)$, $(u'v')$ and the density $p$.

**Vertical Shear:** we now move on to the vertical shear transformation. Note that this transformation can be seen as a special case of a translate but with a translation coefficient varying with row/columns. We obtain the following derivations

$$\mathbb{E}_{\theta \sim \Theta}[T(u,v;\theta)] = \int_{\theta} p(\theta) \int I(x,y)h_{\theta}(u,v,x,y)dxdyd\theta$$

$$= \int_{\theta} p(\theta) \int I(x,y)\delta(u = x + \theta * v, v = y)dxdyd\theta$$

$$= \int I(x,y) \int_{-\infty}^{\infty} p(\theta)\delta(u = x + \theta * v, v = y)d\theta$$

$$= \int I(x,v)p(\frac{u-x}{v})dx$$

$$\implies \mathbb{E}_{\theta \sim \Theta}[T(.,v;\theta)] = I(.,v) \star p(\frac{.}{v}),$$

as a result we see that an efficient way to obtain the expected image (each row/column of it) is via a 1-dimensional convolution with the density being rescaled based on the considered row/column. We now consider the second order moment below

$$\mathbb{E}_{\theta \sim \Theta}[T(u,v;\theta)T(u',v';\theta)] = \int I(x,y)I(x',y')\int_{-\infty}^{\infty} p(\theta)$$

$$\times \ \delta(u = x + \theta * v, v = y)\delta(u' = x' + \theta * v', v' = y')dxdydx'dy'd\theta$$

$$= \int I(x,v)I(x',v')\int_{-\infty}^{\infty} p(\theta)\delta(u = x + \theta * v)\delta(u' = x' + \theta * v')dxdx'd\theta$$

$$= \int I(x,v)I(x',v')p(\frac{u-x}{v})\delta(\frac{u-x}{v} = \frac{u'-x'}{v'})dxdx'$$

$$= \int I(x,v)I(x',v')p(\frac{u-x}{v})\delta(x = u + (x'-u')\frac{v}{v'})dxdx'$$

$$= \int I(u + (x'-u')\frac{v}{v'},v)I(x',v')p(\frac{u'-x'}{v'})dx'$$

$$= \int I(u + zv,v)I(u' + zv',v')p(-z)v'dz,$$

concluding the shearing transformation results. For the horizontal shear, simply do the above derivations with the image axes swapped.

**Rotation:**

$$\mathbb{E}_{\theta \sim \Theta}[T(u, v; \theta)] = \int_{\theta} p(\theta) \int I(x, y) h_{\theta}(u, v, x, y) dxdyd\theta$$

$$= \int_{\theta} p(\theta) \int I(x, y) \delta(u = \cos(\theta)x - \sin(\theta)y, v = \sin(\theta)x + \cos(\theta)y) dxdyd\theta$$

$$= \int_{\theta} p(\theta) \int I(x, y) \delta(x^2 + y^2 = u^2 + v^2, \theta = \arctan(y/x) - \arctan(v/u)) dxdyd\theta$$

$$= \int I(x, y) \int_{\theta} p(\theta) \delta(x^2 + y^2 = u^2 + v^2, \theta = \arctan(y/x) - \arctan(v/u)) dxdyd\theta$$

$$= \int I(x, y) p(\arctan(y/x) - \arctan(v/u)) \delta(x^2 + y^2 = u^2 + v^2) dxdy$$

$$\mathbb{E}_{\theta \sim \Theta}[T(u, v; \theta) T(u', v'; \theta)] = \int_{\theta} p(\theta) \int I(x, y) h_{\theta}(u, v, x, y) dxdy$$

$$\times \int I(x', y') h_{\theta}(u', v', x', y') dx'dy'd\theta$$

$$= \int I(x, y) I(x', y') \int_{0}^{2\pi} p(\arctan(y/x) - \arctan(v/u))$$

$$\times \delta(\arctan(y/x) - \arctan(v/u) = \arctan(y'/x') - \arctan(v'/u'),$$

$$x^2 + y^2 = u^2 + v^2, x'^2 + y'^2 = u'^2 + v'^2) dxdydx'dy'$$

This concludes our derivations. Note that while we focused here on the most common transformations, the same recipe can be employed to obtain the expected transformed image for more complicated transformations.