# OpenReview forum: "A Data-Augmentation Is Worth A Thousand Samples: Analytical Moments And Sampling-Free Training"
_NeurIPS.cc/2022/Conference — NeurIPS 2022 Accept_

### Official Review · Reviewer_97Wc · 2022-07-03

**Rating:** 7
**Confidence:** 3
**Soundness:** 3 good
**Presentation:** 3 good
**Contribution:** 3 good

**Summary:**

The paper aims to provide a theoretical framework for analyzing the role of data augmentations (DA) for learning tasks involving both linear and linear models. To this end, it first proposes a framework by which image transformations are modeled as matrix-vector multiplications using data-space transforms.

This framework is subsequently used to derive the analytical form of the first two moments of an augmented sample. Next, the MSE loss under a linear model is considered in which an exact form of an explicit DA regularizer is derived. An exact expression of the optimal solution under infinite DA sampling is derived, which generalizes the standard Least-Squares solution which is known when no DA is applied. The expression depends exactly on the expectation and variance of transformed samples. For the non-linear case, it is mentioned that the Taylor approximation can generalize from the linear setting.

Next, the paper analyses DA's sampling efficiency and loss sensitivity. An empirical evaluation shows the DA's sampling strategy is inefficient and requires many samples to converge to a good solution. The sensitivity of a given loss and model under a DA is analyzed leading to a derivation of a popular regularization strategy of TangentProp.

**Questions:**

1. How do the derivation provided in Sec.3.4 and 4.2 change assuming a non-linear model? In particular for which range of transformations is the approximation sufficiently close to the linear case?
2. How does the analysis provided in Sec.4.1 generalizes for different transformations and images?

**Limitations:**

The authors mention the limitation regarding the third-order approximation of non-linear models. However, this is not probably characterized. In which cases (range of transformations, images) is it ok to make such assumptions?

**Strengths And Weaknesses:**

Strengths:

1. The paper proposes a novel set of mathematical tools for analyzing DA in a theoretical setting, which has thus been mainly analyzed through an experimental lens. These tools could be important in analyzing proposed DA schemes and so are an important and original contribution.
2. The authors use these tools to derive an explicit regularizer induced by each DA as well as derive the sensitivity, of a given loss and model under a DA policy. These are useful properties for understanding the role of DA as a regularization.
3. The analysis of the variance (sensitivity) leads to a first-principles derivation of a TangentProp as a regularizer which minimizes the loss variance. This is significant as it demonstrates the applicability of the framework in deriving and understanding useful regularizers.

The paper is mostly clear, however many of the proofs are left for supplementary, and at least an intuition could be useful to include in the main text.

Weaknesses:
1. The analysis provided in Sec.4.1 is given for a single image and for a particular transformation. It is unclear if this holds for a more general case and in particular for a wide range of natural images and transformations. For a more general setting, can an expression in terms of the number of images n be given in order to closely approximate estimate the MSE loss from the augmented samples?
2. The analysis provided in the paper in Sec.3.4 and Sec.4.2 is exact for linear models. It is unclear how the derivation is affected under non-linear models even for small transformations. Further, in practice, one often performs large transformations (e.g. in SSL models) which are crucial to their success. Hence it is unclear if the framework provided in this paper can explain the non-linear or large transformation setting well, limiting the significance of the work.

****
Typo, line 176, Gaussian blue

---

> ### Author Response · Authors · 2022-08-02
> **Answer to Reviewer 97Wc**
>
> We thank the reviewer for the positive review and for providing some insightful comments, especially around section 4.
>
> All the analysis from Sections 4.1, 4.2 and 4.3 work the same regardless of the underlying data-augmentation i.e. it does not need to be among the ones that we derived analytically in the paper. This is one of the great benefit of TangentProp (Section 4.3) in that it offers a practical and computationally cheap solution to minimizer the variance of loss under data-augmentation sampling. Regarding the single sample versus dataset analysis, this is a very interesting observation. If we assume truly iid samples, we can perfectly use the analysis from section 4 for the entire dataset as their variance then decompose into the per-sample summation (by independence) and thus all the analysis of section 4 can be ported as is, simply adding a summation over the sample outside of the found expressions. In the non-independent setting however, one would have to compute the cross-covariance terms which could be more intricate. This is a very interesting future direction however with many potential benefits that we would be happy to explicitly mention at the beginning of section 4 if the reviewer finds the above useful.
>
> In the nonlinear regime, all of our results rely on a third order Taylor approximation. This approximation might seem quite trivial and simple, but it is in fact commonly employed in theoretical analysis of deep network as closed-form solution rarely exist without it. Additionally, this approximation has been shown to be surprisingly good enough so that any theoretical insight obtained form it transfers to the actual nonlinear regime (see sec A.2 of [1]).
>
> We entirely agree with the reviewer that one of the main theoretical limitation of our submission is in dealing with large transformations with nonlinear models. As of now, we have an exact characterization of small/large transformations in the linear setting, and only small transformations in the nonlinear settings (although with a third order approximation). We will be sure to reinforce this limitation in the conclusion of our submission. We will also be sure to add intuition into the proofs of each result in the main text, as the camera ready version allows for an extra page.
>
> We remain available to discuss the above point during the discussion period and hope that those changes are satisfactory to the reviewer.
>
> [1] Wei, Colin, Sham Kakade, and Tengyu Ma. "The implicit and explicit regularization effects of dropout." International conference on machine learning. PMLR, 2020.

---

> > ### Comment · Reviewer_97Wc · 2022-08-09
> > **Response to authors**
> >
> > Thank you for providing additional explanations and clarifications. Having read the comments from other reviewers, I believe the experiments going beyond MNIST are important to demonstrate the applicability of the proposed approach. I believe the theoretical foundations provided in the paper are strong and the paper is worthy of acceptance.

---

> > > ### Author Response · Authors · 2022-08-09
> > > **Answer to Reviewer 97Wc**
> > >
> > > We are thankful to the reviewer for their answer, and for their appreciation of our submission.
> > > We also entirely agree with the reviewer on the value added to our submission by running those experiments, and we are glad that those updates and our comment have addressed the reviewer's original concerns.

---

### Official Review · Reviewer_JxCU · 2022-07-10

**Rating:** 7
**Confidence:** 3
**Soundness:** 3 good
**Presentation:** 3 good
**Contribution:** 3 good

**Summary:**

The authors propose a novel operator called Data-Space Transform (DST) to express data augmentation (DA) and use it to investigate three topics: 1) equivalent regularizer induced by each DA; 2) the number of DA samples for a Monte-Carlo (MC) approach to correctly estimate the information of the DA; 3) sensitivity analysis of loss/model pairs under a DA policy. The third topic led to a rediscovery of TangentProp from first principles.

**Questions:**

Section 4 (as well as the title)  suggests that the simple translation augmentation needs around 1000 to 10000 samples to correctly estimate the MSE loss. In the 2012 Alexnet paper, by counting the number of unique (fixed-sized) crops of sampling a 224x224 patch from a 256x256 image, a 1024x gain in the number of training samples is estimated. I understand that this work investigates the problem in a different framework, but can the authors comment whether these two estimates are related in some way?

**Limitations:**

The authors discuss various limitations of this work in several sections which include important mathematical assumptions and approximations. Overall I believe the limitations are well discussed.

**Strengths And Weaknesses:**

Strengths:

1. The paper is generally well written and motivated.

2. The new transform operator proposed is intuitive.

3. One of the key discoveries enabled by DST is a closed-form expected loss for DAs (for linear models and MSE loss), which allows for measuring the number of samples needed to correctly estimate the information conveyed by each DA.

4. The rediscovery of TangentProp from principled derivation provides a nice link to practical use cases beyond the linear model and MSE loss case.

5. Limitations of the work including various assumptions and approximations are discussed in the paper.

Weaknesses:

Major:

1. As shown in figure 5, it seems that even for the linear model and MSE loss combination where an analytical solution is available, the classic MC sampling DA method can outperform the expected loss method with large enough training sets (e.g. N=3000 for MNIST), which suggests that there are aspects not captured by the closed-form loss. Although it’s hard to evaluate whether some of the effects as error bars are not included in this figure, the trend shows that increasing the size of the training set could lead to larger performance lead for the multi-sample DA method over the expected loss approach. Can the authors 1) provide error bars, 2) comment on why MC DA may outperform the expected loss method for the linear model, and 3) conduct experiments with larger training sets?

Minor:

1. Results for non-linear models seem to be limited to small augmentations. Many state-of-the-art augmentations are rather aggressive, how well does the results generalize to these aggressive augmentations?

2. Figure 3 caption, missing space before “Fourth column”

3. Line 243: “than” => “as”

4. Line 263: remove “as”

---

> ### Author Response · Authors · 2022-08-02
> **Answer to Reviewer JxCU**
>
> We are grateful for the reviewer's positive review and for pointing out that we correctly mentioned the assumptions and the limitations of our work in the main text as part of our derivations.
>
> We will be happy to add a table summarizing the average and std of the performance depicted in figure 5 to provide improved readability of those results (which is indeed poorly conveyed from the figure). We thank you for this suggestion. Regarding the possible gain of using MC DA versus the closed form solution, this is an interesting observation. One possible explanation might be that in some realization of the random DA sampling, the actual distribution of the DA parameters are quite far from the analytical distribution used in practice (this would go away as the number of augmented samples grows to infinity). Hence one possible reason could be that the empirical DA parameter distribution and the analytical one are slightly different, and that this produce a slightly different effect on the model's parameters that in turn translate into possible better performances. This is an interesting observation that we will be sure to comment on as part of the figure caption. We will be happy to provide experiment with larger training set: in short, the benefit of DA quickly saturates and all the curves (true expectation vs varying number of DA samples) collide into one (as expected).
>
> In the nonlinear regime, all of our results rely on a third order Taylor approximation. This approximation might seem quite trivial and simple, but it is in fact commonly employed in theoretical analysis of deep network as closed-form solution rarely exist without it. Additionally, this approximation has been shown to be surprisingly good enough so that any theoretical insight obtained form it transfers to the actual nonlinear regime (see sec A.2 of [1]). Yet, we agree with the reviewer that one of the main theoretical limitation of our submission is in dealing with large transformations with nonlinear models. We will be sure to reinforce this limitation in the conclusion of our submission.
>
> We thank the reviewer for pointing out to the typos in our manuscript, we will be sure to correct them. We remain available to discuss the above point during the discussion period and hope that those changes are satisfactory to the reviewer.
>
> [1] Wei, Colin, Sham Kakade, and Tengyu Ma. "The implicit and explicit regularization effects of dropout." International conference on machine learning. PMLR, 2020.

---

> > ### Comment · Reviewer_JxCU · 2022-08-08
> > **Response to authors**
> >
> > Thanks for providing explanations as well as updating experiments. I believe that this paper is generally strong but it can be further improved with experiments beyond MNIST as pointed out by reviewer yaYp and reflected by my rating.

---

> > > ### Author Response · Authors · 2022-08-08
> > > **Answer to Reviewer JxCU**
> > >
> > > We thank the reviewer for providing great feedbacks and pointing out that the reviewer finds value in our submission. We agree with your and Reviewer yaYp's concerns and have added sets of experiments on FashionMNIST and EMNIST (we kindly point the reviewer to our latest general answer summarizing those plus the gained insights). We believe that those added experiments will allow to obtain not only theoretical but also practical insights into the impact of DA.
> > >
> > > We believe that this addresses the reviewer's concern but we remain happy to answer any additional question or concern that the reviewer might have.

---

### Official Review · Reviewer_nBxs · 2022-07-12

**Rating:** 7
**Confidence:** 3
**Soundness:** 4 excellent
**Presentation:** 3 good
**Contribution:** 3 good

**Summary:**

The paper provides a theoretical analysis of data augmentation (DA) methods.
- It introduces a novel integral transform formulation (called Data-Space Transform (DST)) for describing a class of DA, e.g. translation or rotation.
- It provides an explicite formula for expectation and variance of transformed images
- It provides an explicite formula for regularisation that has the same effect on the loss as training with infinite DA samples
- It provides a theoretical connection with TangentProp


**Questions:**


Does the method help reduce the number of steps to train a model in a non-linear case?

Does it help generalisation in the regime with small data-set in the non-linear case? Fig 7 did not show results with explicit regularisation.

Please explain the experiments on fig 5 and 7: what is the exact loss, training set, number of epochs and relevant hyperparameters for each scenario.

**Limitations:**


The authors addressed some limitations (see weaknesses) except for the efficiency of the method.

Please discuss how the explicate regularisation can be computed efficiently, i.e the spectral decomposition. Without this the method cannot work on large images.

**Strengths And Weaknesses:**

Strengths:

The paper provides some insights of the effect of DA:
- How it has the same effect as an explicite regulariser
- It can explicitly recover the expectation and variance of the transformed samples
- It provides interesting connections between the variance of the MC estimate and the model’s Jacobian matrix, which leads to the TangentProp regulariser (that minimises the variance).

The theoretical analysis looks original to me.

The paper discusses a very important topic, as understanding the effects of DA and regularisation could help design systems that generalise better.

Weaknesses:

The presentation could be clearer:
- fig 1 does not help understanding. One has to zoom a lot to see the details. I had to actually read the text to understand the image. Perhaps a much lover resolution image (6x6 max) depicting something very simple like a circle would be better. Then the reader could understand the transformation matrix. Also there should be an indication in the figure how the image is flattened.
- figures are referenced out of place, eg. fig 1 in page 5 and some in captions in other figures.
- It is not clear how the linear model for MNIST in fig 5 was trained. I am assuming. The training set is created with N DA samples. For all N, the number of epoch are the same. For the black line the explicite regularisation is used.

The proposed method has many limitations:
- The class of DA is limited to the cases where they can be expressed as a linear function on images. Many relevant DA falls outside of this (e.g. jpeg compression).
- It is hard to use in practice, as deriving the expectation and variance of a bit more complicated DA (e.g. some non-linear warp) is a very had manual task.
- For non-linear models the method relies on 3rd order Taylor expansion.
- Even in the linear regression case, for a bit complicated DA the method requires the spectral decomposition of a very large system (a matrix of size n^2 x n^2). It is not discussed how this computation can be done efficiently (I assume it can be done for simple DA). If it cannot be done, then what are the trade-offs?

It is not clear to me how practical the explicite regulariser is. See Questions.

Overall I rate the significance of the method not very high because of the limitations, that I think are severe.

EDIT:
The rebuttal changed my mind, I think the paper is worth reading despite the limitations of the method in its current form. The significance is in the theory and that is valuable.

---

> ### Author Response · Authors · 2022-08-02
> **Answer to Reviewer nBxs**
>
> We thank the reviewer for their positive comments, especially for acknowledging the originality of our theoretical analysis, and for pointing out the importance of the conclusions we have reached (linking data-augmentation and regularization).
>
> You are correct regarding the experimental setup of Fig.  5. We will be sure to better introduce those in the caption, and we would also like to take this opportunity to remind the reviewer that we will be releasing all the codebase to reproduce the figures/experiments upon final decision. We will also carefully edit the figure references and improve Fig. 1 readability by providing a greater zoom factor on the bounding boxes of the sparse matrix.
>
> Regarding the other comments, we believe that our general comment answered most of your concerns. We entirely agree with the reviewer with the practical limitations of the derived regularizer (very large matrix to be computed, although sparse, this would not be a practical replacement to performing data-augmentation directly). But we hope that the theoretical insights and the ability to use those insights to then derive further methods (e.g. TangentProp and possible variations of it) outweighs those limitations.
>
> Regarding your questions, we believe that there would not be a practical benefit of employing the regularizer explicitly for a nonlinear model for two reasons. First, in the nonlinear model setting, the regularizer is only valid for small transformations as it relies on a third order Taylor approximation (as well noted by the reviewer) and it is not clear if deep networks benefit greatly from such small transforms. Second, the cost of performing data-augmentation is relatively small compared to the cost of a forward-backward pass in a network. Hence our alternative would not provide any benefit in that regard (even if computation of the explicit regularizer was fast and optimized). If those answer are agreeable by the reviewer, we would be delighted to mention them in the conclusion/limitation section to better help the reader understand the scope of our results.
>
> We remain available to discuss the above point during the discussion period and hope that those changes are satisfactory to the reviewer. We believe that they will make our submission stronger and more focused on our core contributions.

---

> ### Comment · Reviewer_nBxs · 2022-08-09
> **rebuttal well received**
>
>
> "we provide for the first time a thorough demonstration that employing data-augmentation during training of a model, corresponds to employing no data-augmentation but an explicit regularizer, and that we demonstrate how to obtain this regularizer if one aimed at inspecting (visually or statistically) the properties of that regularizer and how it impacts a model's parameters."
>
> I thank for the authors for their rebuttal and clarifications. I think showing a theoretical connection like this is valuable and makes the paper worth reading for people who seek more theoretical understanding.
>
> Even if the method is not (yet) practical to be deployed in real world tasks, it may be a seed idea for future breakthroughs. I also appreciate that the authors are open about this and discuss the limitations of the method openly.
>
> Under this light I can see that the experiments were aimed at testing the (scientific) claims of the paper and not to break new ground in performance on some large benchmark X. In my opinion the experimental results are sufficient for this.
>
> The authors thoroughly answered my questions as well, thus I increase my rating.

---

### Official Review · Reviewer_yaYp · 2022-07-13

**Rating:** 4
**Confidence:** 3
**Soundness:** 2 fair
**Presentation:** 2 fair
**Contribution:** 2 fair

**Summary:**

The authors study both theoritically and empirically the sample efficiency of training with data augmented samples on a set of augmentations such as translation, shearing, rotation and zoom. They derive mathematical tools for analytically studying data augmentations and its effects on loss. As a result they derive an explicit regularizer that produces a model that is the equivalent of using DA on non-trivial transformations (the first of its kind).

The authors also study the number of DA samples needed for a model to properly absorb the information given by a data augmentation strategy on a particular image, as well as the number of DA samples needed in a setting where we have thousands of unique images for a model to fully absorb such information. They find that for a single image that number is close to 10 ^ 4, and for datasets with thousands of images, it can be around 50 data augmented versions of each image.

The authors also study the variance of a given loss and architecture applied on a particular data augmentation strategy and rediscover a popular reqularizer called TangentProp.

**Questions:**

I recommend the following:
- Repeat your experiments on more datasets and models to see whether any of the conclusions related to them hold.
- Better motivate/communicate what your findings mean for future research and/or current machine learning practices.
- Rewrite parts of your paper with an aim towards more clarity and more conciseness.

**Ethics Review Area:**

["I don’t know"]

**Limitations:**

Technical limitations were discussed above. I don't see any potential negative societal impacts.

**Strengths And Weaknesses:**

Originality:

The investigation is both interesting and insightful, and follows a novel direction in how it conducts an analytical study on data augmentations and attempts its best to come up with a theoretical framework for studying the effects of data augmentation on models and losses.

Quality:

The technical rigour of the work is of high quality, but the writing can vary in quality, sometimes making it harder to follow what the authors are trying to communicate.

Experiments are strictly done on MNIST, which is a majour concern as to whether any of the empirically supported conclusions hold.

Clarity:

The writing quality of the paper is mediocre, but the authors have definitely excelled in their mathematical communication in the paper. Furthermore, the figures presented would benefit from additional textual descriptions of what they portray.

Significance:

The conclusions regarding the sample efficiency of data augmentation strategies can be useful for people in the field if applied on more datasets and models. Furthermore, I am not sure how TangentProp can help improve machine learning models in any additional ways as a result of this study.

The significance of the study is quite uncertain.

---

> ### Author Response · Authors · 2022-08-02
> **Answer to Reviewer yaYp**
>
> We thank the reviewer for their positive comments, especially noticing the technical rigor of our submission and the importance of the conclusions we have reached.
>
> We believe that our general comment mostly answer the reviewer's concern. We simply would like to briefly repeat that as we do not aim to provide a practical solution to replace the use of data-augmentation, we focused on providing a thorough theoretical analysis with as many insights as possibles (e.g. the possible link between translation and random crop DA and a model focusing on texture as opposed to shapes). Hence we believed that empirical validation on MNIST was sufficient to carry a proof of concept experiment.
>
> We entirely agree with the reviewer that the figures would benefit from additional details in their caption (and when they are introduced in their corresponding paragraphs). We will be sure to add those for the final revision as permitted by the additional page for the camera ready version.
>
> We will also be sure to add a limitation section as part of the conclusions summarizing our general comment (on the fact that we do not provide a practical alternative to data-augmentation).
>
> We remain available to answer any further question the reviewer might have on the above points during the discussion period.

---

> > ### Comment · Reviewer_yaYp · 2022-08-07
> > **Response to rebuttal**
> >
> > Thank you for your response.
> >
> > >We believe that our general comment mostly answer the reviewer's concern. We simply would like to briefly repeat that as we do not aim >to provide a practical solution to replace the use of data-augmentation, we focused on providing a thorough theoretical analysis with as >many insights as possibles (e.g. the possible link between translation and random crop DA and a model focusing on texture as opposed >to shapes). Hence we believed that empirical validation on MNIST was sufficient to carry a proof of concept experiment.
> >
> > I agree with you until you said that MNIST was sufficient. MNIST holds zero to no information for most modern deep nets. In all truth, MNIST is so bad as a benchmark dataset for drawing any conclusions related to deep nets that literally anything applied on it will return the same exact score, that is, a perfect accuracy score in evaluation. Something like EMNIST, or Omniglot would be the bare minimum for me to be able to consider accepting this paper.

---

> > > ### Author Response · Authors · 2022-08-08
> > > **Answer to Reviewer yaYp**
> > >
> > > We thank the reviewer for providing those useful feedbacks. We entirely align with the reviewers' comments and have thus added (as suggested) two set of experiments on Fashion-MNIST and EMNIST. We refer the reviewer to our general answer for additional details and we believe that this addresses the reviewer's concern on the lack of empirical use for our submission.
> > >
> > >
> > > We remain happy to answer any additional question or concern that the reviewer might have.

---

> > > > ### Comment · Reviewer_yaYp · 2022-08-09
> > > > **Nice results -- more clarifications and paper reformatting needed**
> > > >
> > > > Thank you for running those experiments!
> > > >
> > > > I like the results, however the table naming of the various hyperparameters is a bit unintuitive. You use ds=N for dataset size, and then dsxN to indicate the size of the data augmentation. Just use something like N vs M and clearly state below or in a legend what each means. As it currently stands it's easy to misinterpret.
> > > >
> > > > Also, the new results need to be incorporated into the base paper, and integrated into the discussion section. If that is done, I am willing to accept this paper with flying colours :)

---

> > > > > ### Author Response · Authors · 2022-08-09
> > > > > **Answer to Reviewer yaYp**
> > > > >
> > > > > We first would like to thank the reviewer for the forth and back discussion. We believe that this has allowed us to greatly improve our submission by adding additional experiments (previous revision) and now to improve their formatting (latest revision).
> > > > >
> > > > > In the event of acceptance, we are allowed for an additional page in the main part of the submission (not during rebuttal). Yet, to be sure that the reviewer sees precisely what we intend to bring into that extra page, we have added in the supplementary material the *final formatting* of the table summarizing those experiments (**Appendix A, Table 1**). This table (and its caption) will be the one going into the main part of the paper and it will obviously be well introduced in the experiment sections (discussions included in the current caption).
> > > > >
> > > > > We have also made sure to **improve the notations in that table**, if the reviewer agrees that this formatting is simpler to read, it will naturally be adopted for the appendix tables as well. We hope that this addresses the reviewer's latest comment and remain available to answer any further question.

---

### Author Response · Authors · 2022-08-02
**General answer and clarification of our main contribution**

We first would like to thank all the reviewers for providing qualitative comments and suggestions. Although each reviewer presented a few specific comments and suggestions (that we will answer in a per-reviewer comment), we identified two major themes among most reviewers which were also the main weaknesses of the paper that those reviewers have put forward:

1. how does our method can help practitioners directly i.e. how to deploy the derived analytical regularizer that we obtained to further improve current models' performances
2. how to include more recent data-augmentations that enabled state-of-the-art performances in deep learning into our analysis

However, we were careful to never mention in the abstract/introduction such application of our method since (and as well pointed out by the reviewers) we do not believe that our analytical regularizer can provide **in general** a practical and scalable alternative to performing data-augmentation directly. Instead, if we had to summarize the contribution of our paper in a single point, it would be that **we provide for the first time a thorough demonstration that employing data-augmentation during training of a model, corresponds to employing no data-augmentation but an explicit regularizer, and that we demonstrate how to obtain this regularizer if one aimed at inspecting (visually or statistically) the properties of that regularizer and how it impacts a model's parameters.**

As a byproduct of our analysis, we also show that one possible practical solution to reducing the variance of the loss under DA sampling (without employing the DA regularizer explicitly) is to employ TangentProp, **which has already been introduced and successfully employed for years, and would thus not be part of our practical contributions**.

Despite the lack of practical deployment of our methodology, we believe that the tools we have introduce and the theoretical conclusions we obtained are of crucial interest to the community. Here is an illustrative example that we would be happy to add to our final revision if the reviewers agree on its benefits:

- suppose that one was to employ data-augmentation only on a subset of the training set e.g. to fight some spurious correlation or model bias, as recommended by [1,2]
- as per our analysis, doing so simply amounts to employing the same data-augmentation regularizer but simply with a lower amplitude i.e. the amount of regularization is lessen but the form of the regularizer is the same
- hence, we can conclude that the exact same effect (beneficial or not) could be obtained by simply applying the data-augmentation with a random probability to be "on/off" **regardless of the training sample it is applied onto** and thus without the need to first identify the subgroups in a given training set that suffer or not from the application of the data-augmentation.

The above alone could for example speed up the deployment of such methods. There are of course many additional such applications of the insights we obtained here. Hence, we believe that a lack of direct practical value does not reduce the value of our submission. Beyond such cases, we believe that the connection between data-augmentation and explicit regularization for non-trivial policies (e.g. rotation, translation, shearing, etc) that we studied in our submission, should be insightful for a large part of our community. Although those augmentations might look trivial for practitioners, from a theoretical standpoint, very little is known beyond dropout, and thus including those policies into the set of existing theoretical results seems quite relevant.

**We will be sure to answer any possible question that the reviewers might have during the discussion process to be sure that we can update the writing of our submission to ensure that no reader expects a practical solution from our analysis.**

[1] Sharma, Shubham, et al. "Data augmentation for discrimination prevention and bias disambiguation." Proceedings of the AAAI/ACM Conference on AI, Ethics, and Society. 2020.

[2] Iosifidis, Vasileios, and Eirini Ntoutsi. "Dealing with bias via data augmentation in supervised learning scenarios." Jo Bates Paul D. Clough Robert Jäschke 24 (2018).

---

### Author Response · Authors · 2022-08-04
**Summary of edits done to the revised submission**

We would like to thank once again all the reviewers for their insightful comments and for their suggestions. We have implemented them and we feel that our submission has been strengthen as a result. **We also reiterate here that all the codebase to reproduce any figure/table will be publicly released upon completion of the review process hoping that future work can be built on top of ours**. Here is a brief summary of changes that we have performed (*all major edits are in blue in the paper*):

- *clarified the application scope of the paper*: we have added a sentence in the introduction further reinforcing that we are not proposing a tractable alternative to data-augmentation, but rather a comprehensive theoretical study that could have the potential to bring insights for understanding how does DA impacts the parameters of a model at hand
- *added four empirical result tables with standard deviation over multiple runs/DA settings*: as well suggested by a few of the reviewers, we have added four tables in the appendix with the exact average +- std classification accuracy for many settings of DA, dataset size, and number of DA samples (to compare with the given result using the analytical regularizer we obtained). We hope that all those settings will reassure the reviewers on the empirical results that were originally proposed in the paper
- *corrected typos and figure references*
- *added the experimental details in the appendix for the original convergence figures*

We hope that those changes satisfy the reviewers and we remain available for further discussions.

---

### Author Response · Authors · 2022-08-08
**Additional Experiments on EMNIST and FashionMNIST with avg/std**

We are grateful to the reviewers for providing feedbacks during the discussion process. As per the current reviews, it seems that all concerns have been addressed except for the lack of more complex empirical validations (we remain available throughout the discussion process if some concerns remain unaddressed or if new comments appear to the reviewers).

To that end, and as well suggested by **Reviewers yaYp and JxCU** we have added experiments on EMNIST (**Table 5 to 8**) and FashionMNIST (**Table 9 to 12**) to complement the MNIST (**Table 1 to 4**) experiments, all with average and std over 10 runs. In particular we provide those in the appendix as table where we vary the dataset size (up to 20,000 as per question) and for four different DA pipelines (combination of rotation, translations and scaling).

*In the event of acceptance, one table for each dataset will be move to the extra page with a more compact formatting and with discussions regarding the following points...*

We obtain two very interesting insights:

1. **in the large dataset size regime** the gap between low number of DA samples and the expected regularizer is proportional to how accurate is the DA to encode the class invariances. This follows preliminary observations, if we have enough data samples that already contain variations as would be expressed from a given DA policy, using many DA samples or the expected regularization only provides marginal gains
2. **in the small dataset size regime** a perhaps more surprising finding is that even if the DA policy is not well aligned with the class invariances (e.g. rotation for fashion MNIST) there is still a dramatic gain of using the regularizer simply to prevent the otherwise extremely poor solution obtained by the usual weight decay regularization (without any, no unique least square solutions exists as there are less samples than number of parameters). Hence we reach the surprising conclusion that *even if a DA is misaligned with the task at hand, in the very low data regime it will still boost performance by improving the conditioning of the loss landscape*.

We hope that those findings provide additional empirical value to our submission and that we were able to address the reviewers' concerns.

---

### Meta-Review · Area_Chair_Shmk · 2022-08-26

**Recommendation:** Accept
**Confidence:** Certain

**Metareview:**

This paper studies data augmentation through the lens of an explicit regularizer, deriving a closed form regularization term corresponding to the effect of data augmentation in the case of linear models and MSE, and analyzes its properties with respect to convergence, sample efficiency and stability. Data augmentation is a core technique in deep learning that is poorly understood and has not, to my knowledge, been the subject of much rigorous analysis, so this work has the potential to be quite influential in our understanding of a fundamental deep learning practice.

Reviewers were generally positive on the approach and the theoretical underpinnings. Several were concerned about generalization of the approach to nonlinear models, which the authors explain proceeds by means of a Taylor expansion. The main shortcoming according to reviewers was experiments limited to MNIST, which the authors remedied with extension of their experiments to additional datasets (EMNIST and Fashion MNIST, to be precise).

With scores uniformly deep within acceptance territory with the exception of yaYp who promises to "accept this paper with flying colours" if the new results are incorporated into the main paper, this seems like an obvious candidate for acceptance.

P.S. The technique of deriving a closed form penalty is reminiscent of marginalized dropout (Wager et al, 2013), a work I would suggest deserves citation in the camera ready.

**Award:**

Yes

---

### Decision · Program_Chairs · 2022-09-14

Accept